

# Zonal scale and temporal variability of the Asian monsoon anticyclone in an idealised numerical model

Philip Rupp[1,2] and Peter Haynes[1]

[1]Department of Applied Mathematics and Theoretical Physics, University of Cambridge, Cambridge, United Kingdom
[2]Current affiliation: Meteorological Institute Munich, Ludwig-Maximilians-University, Munich, Germany

**Correspondence:** Philip Rupp (philip.rupp@lmu.de)

**Abstract.**

The upper-level monsoon anticyclone is studied in a 3-D dry dynamical model as the response of a background circulation without any imposed zonal structure to a steady imposed zonally confined heat source. The characteristics of the background circulation are determined by thermal relaxation towards a simple meridionally varying state, which gives rise to baroclinic instability if meridional gradients are sufficiently large. This model configuration allows study of the dependence of the monsoon anticyclone response on characteristics of both the imposed heating and the background state, in particular including interactions between the anticyclone and the active dynamics on its poleward side in the form of the jet and baroclinic eddies.

As characteristics of forcing and background state are varied a range of different behaviours emerges, many of which strongly resemble phenomena and features associated with the monsoon anticyclone as observed in re-analysis data. For a resting background state the time-mean anticyclone is highly extended in longitude to the west of the forcing region. When the active mid-latitude dynamics is included the zonal extent of the time-mean anticyclone is limited, without any need for the explicit upper-level momentum dissipation which is often included in simple theoretical models, but difficult to justify physically.

We further describe in detail the spontaneous emergence of temporal variability in the form of westward eddy shedding from the monsoon anticyclone for varying strength of the imposed heating. By varying the strength of the background mid-latitude dynamics we observe a transition of the system from a state with periodic westward eddy shedding to a state dominated by eastward shedding. The details of the time-mean structure and temporal evolution depend on the structure of the background flow and for certain flows the monsoon anticyclone shows signs of both westward and eastward shedding.

## 1 Introduction

### 1.1 The monsoon circulation

One of the major features of the atmospheric general circulation is the Asian monsoon. This seasonal large scale circulation pattern is active during the boreal summer months, peaking during June-August (JJA), and spans large parts of south and south-east Asia. The monsoon can strongly influence the weather and climate of the northern hemisphere (NH) via various mechanisms, e.g. locally through strong convective rainfall in the monsoon region or globally since it potentially forms an





important transport pathway from the troposphere into the stratosphere and can thus affect the state of the stratosphere. In the lower and middle troposphere the characteristics of the monsoon are dominated by large-scale convection and precipitation. Predicting the corresponding monsoon rainfall is crucial, e.g. to maximise the agricultural output within the monsoon region, making it essential to minimise the uncertainties associated with the monsoon and its dynamics.

The lower and middle tropospheric circulation of the monsoon is characterised by cyclonic flow, while in the upper tropo-
sphere and lower stratosphere (UTLS) the monsoon is characterised by a large anticyclone. Many properties of the large-scale dynamics of the monsoon flow can be explained in terms of the forcing of potential vorticity (PV) that is provided by the latent heating due to the convective precipitation. The latent heating maximises in the middle troposphere, providing a positive PV forcing below and a negative PV forcing above the maximum, and hence to corresponding cyclonic and anticyclonic circulations. The lower and upper level circulation of the monsoon are linked via large-scale convective transport and, in combination,
potentially form a fast pathway for air masses (and with it chemical species and aerosols) from the surface into the stratosphere (Randel et al., 2010; Bourassa et al., 2012). Air parcels arriving in the upper troposphere can be transported across the tropopause in two ways, either vertically by the large-scale upwelling of the monsoon or, alternatively, horizontally into the extratropical lower stratosphere through baroclinic mid-latitude eddies (Dethof et al., 1999; Vogel et al., 2014; Ploeger et al., 2013). The latter process depends strongly on the spatial and temporal structure of the monsoon anticyclone, as well as the
dynamical properties of the flow in mid-latitude. The sharp PV gradients that form the edges of the monsoon anticyclone have been suggested to act as a transport barrier that inhibits mixing of air parcels with surrounding air (Ploeger et al., 2015).

## 1.2  Large scale friction in the UTLS and zonal scale of the monsoon anticyclone

The simple steady theory developed by Matsuno (1966) and Gill (1980) for heat-induced tropical flows is widely used to explain the general structure of the monsoon circulation. The Gill-Matsuno model describes the response to a localised, low-
latitude heating in terms of westward propagating Rossby waves and eastward propagating equatorial Kelvin waves, where the importance of the latter strongly depends on the position of the heating relative to the equator. A common critique of the Gill-Matsuno theory is that it requires inclusion of linear friction throughout the depth of the atmosphere in order to obtain a zonally confined response by dissipating the Kelvin and Rossby waves. While there is likely a boundary drag acting on the cyclonic low level flow it is still an open question if any atmospheric mechanism can provide a significant large scale momentum damping
to affect the monsoon anticyclone in the UTLS region (Battisti et al., 1999; Lin et al., 2008; Romps, 2014).

One potentially relevant concept takes account of the vertical transport of horizontal momentum in cumulus clouds, which might provide a large-scale 'cumulus friction'. Zhang and McFarlane (1995) showed that the inclusion of cumulus friction can lead to substantial changes in the overall response of a general circulation model, including effects in the monsoon region. Lin et al. (2008) investigated if the friction used by Gill (1980) can be interpreted as cumulus friction in the UTLS and conclude
that the process can indeed account for a significant damping, although they point out the strong dependence of the resulting cumulus friction damping rate on the strength of the convective systems and the resulting spatial inhomogeneity of the effect. Romps (2014) use a simple model for the vertical convective momentum flux and conclude that cumulus friction can act as a large scale friction with time scales of the order of 1-10days for features of vertical wavelength between 2-10km.





However, the use of cumulus friction as damping mechanism for the monsoon anticyclone is questionable since it is not
fully understood yet to what extent cumulus momentum transport, based on localised vertical cloud movements, can act as a
large scale friction. Different authors have studied the significance of friction in creating the upper level monsoon flow. Holton
and Colton (1972) found that a strong friction is required in order to match their model response to observational data when
the model was linearised about a zonal mean flow. On the other hand, Sardeshmukh and Held (1984) and Sardeshmukh and
Hoskins (1985), who analysed the upper tropospheric vorticity balance for general circulation model output and re-analysis
data, respectively, found a significant contribution from horizontal advection and argued that (vertical) convective cumulus
transport is of lesser importance. They concluded that the dynamics in the tropical upper troposphere is essentially non-linear
and nearly inviscid. Studies such as these are often cited as important, but as yet no clear consensus has emerged on what
physical processes balance the negative forcing of potential vorticity above the monsoon heating and hence control the zonal
structure of the anticyclone.

Other studies used a slightly different approach to obtain a (quasi-)realistic monsoon response in three-dimensional numer-
ical simulations of the tropical middle atmosphere, e.g. Hoskins and Rodwell (1995) or Liu et al. (2007). The general idea in
these studies was to relax the zonal mean zonal wind component towards climatological or idealised profiles, while keeping
the wave part freely evolving. This approach obtained tropical circulations that were localised in longitude. However, fixing
the zonal mean zonal wind tends to suppress non-linear processes as it does not allow for full wave-mean-flow interactions and
it is further not clear whether such an approach is in practise equivalent to some form of large scale friction.

Another approach to modelling the upper level monsoon response to thermal forcing in a quasi-realistic setting has been to
suppress the development of baroclinic instabilities via an imposed damping or to avoid their occurrence by restricting model
integrations in time (usually to about 20 days). An example of this is the work of Hoskins and Jin (1991), who performed
initial value experiments for tropical circulations. Among other cases they perturbed a resting atmosphere and a climatological
zonal mean zonal wind profile with a Gill-like stream function pattern to find the climatological wind profile to significantly
suppress the equatorial flow and causing a polewards propagation of Rossby waves. This study, however, can only give limited
insights into a potential (steady) zonal confinement of the response since the model is not constantly forced and the initial flow
becomes strongly modified due to baroclinicity after about 14 days.

Hoskins and Rodwell (1995) modelled the monsoon circulation as response to thermal or orographic forcing and investigated
the importance of non-linear effects and mountains. They found the general structure of the monsoon anticyclone to be well
reproduced by a linear, thermally forced model. However, again Hoskins and Rodwell (1995) restricted their experiments to
early times in order to avoid baroclinic instabilities developing in their system. Such short integration lengths might simply
bypass the problem of unrealistic zonal extension without the system actually reaching a steady state. Jin and Hoskins (1995)
used a very similar approach and modelled zonally confined Gill-like flow structures as response to an equatorial heating, but
also limited considerations to early times.

Ting and Yu (1998) investigated the steady response of a baroclinic atmosphere with climatological basic state to a localised
equatorial heating, but suppressed the occurrence of baroclinic instability by adding a 15-day mechanical and thermal damping
to the system. Such added friction processes directly affects the explicitly forced flow, which potentially explains the zonal





localisation of the response in their experiments. Hendon (1986) performed Gill-like tropical heating experiments in a 2-layer
model with baroclinically unstable background and an inviscid upper layer to study the relative position of the upper level
anticyclones without restricting the experiments to early times or suppressing the instability of the basic state. However, the
problem of zonal localisation was not addressed in this study in any detail. In Section 3 below we show that baroclinic instability
(which is ignored or suppressed in many of these investigations) can in fact play an important localising effect on the monsoon
circulation.

The problem of zonal confinement of a flow induced by a local heat source was recently investigated by Amemiya and Sato
(2018) in a single-layer model, who found that a meridionally varying depth can have a localising effect. Such a meridional
depth gradient does introduce a zonal mean wind (a westerly jet in their case) and does therefore correspond to a simple
representation of certain dynamic mid-latitude characteristics. However, the basic state used includes regions with negative
meridional PV gradient and thus its relevance for describing the monsoon system is not clear.

## 1.3  Eddy Shedding by the monsoon anticyclone

Alongside characteristics of the time-mean monsoon anticyclone, such as its zonal localisation, the temporal variation of
the anticyclone is also not yet fully understood. The time variation is potentially important because of its meteorological
implications and also, as noted previously, for its implications for transport of chemical species from the upper troposphere,
including from within the centre of the anticyclone, into the extratropical stratosphere. Different authors have observed and
described westward and eastward shedding of vortices from the (bulk) monsoon anticyclone. These vortices (or eddies) are
often defined via localised, coherent PV or geopotential height structures. Fluctuations in flow associated with these vortices
can transport trace gases or water vapour out of the monsoon region (Garny and Randel, 2013; Vogel et al., 2014).

The mechanisms for the two types of time dependence, i.e. westward shedding and eastward shedding, are potentially very
different and the two phenomena have been previously investigated by various other authors. Hsu and Plumb (2000) studied a
particular westward shedding event of the monsoon anticyclone and showed that the response of a single-layer model to a local
steady forcing can become unstable for certain parameter ranges, giving a potential explanation for the westward shedding
process. Indeed Davey and Killworth (1989) had previously analysed the response to a steady mass source in a single layer
beta-plane model intending to describe and explain oceanographic phenomena (e.g. Mediterranean outflow). They observed
a transition from a steady state, a westward extending patch of low PV (sometimes referred to as beta-plume), to an eddy-
shedding state as the strength of the forcing increases, similar to the transition shown later in Figure 11. Davey and Killworth
(1989) further argued that eddy shedding occurs when the meridional PV gradient within the beta-plume of the steady linear
solution becomes negative and the system therefore unstable.

Popovic and Plumb (2001) demonstrated with re-analysis data that such westward shedding events occur regularly and
about 2-4 times each monsoon season. Analysing Fourier spectra of ERA-I PV data Fadnavis et al. (2018) concluded that the
periodic eddy shedding events from the monsoon anticyclone happen with a frequency on synoptic time scales ($\approx$10 days).
Their estimate for the frequency of the almost-periodic shedding process seems to be in rough agreement with the findings of
Popovic and Plumb (2001).





The Liu et al. (2007) work noted previously studied the response of to idealised and semi-realistic monsoon thermal forcing distributions and observed a periodic shedding of eddies from the main PV low of the monsoon anticyclone for sufficiently

strong heating magnitudes. The analysis and model setup used by Liu et al. (2007) mainly differs from the study reported in this paper in two aspects. For one, Liu et al. (2007) emphasise the importance of 'Tibetan heating' in creating the phenomenon of westward eddy shedding, while we will mostly study in detail the changes in temporal variability of the response as the forcing magnitude increases and, in addition, perform a set of experiments in a basic state with temporally varying background flow (Section 4). Second Liu et al. (2007) relaxed their system towards a prescribed steady zonal flow profile which, as mentioned

earlier, can potentially alter the dynamics and restrict the time-mean response. The model configuration used in this study (see Section 2 for details) does not restrict the winds of the forced response.

Rupp and Haynes (2020) found that westward shedding of eddies from a localised region of steady negative PV forcing in a single-layer beta-plane model can be interpreted as result of a spatio-temporal instability of the system. They further analysed the dependence of the underlying instability and the resulting qualitative state of the system on different characteristics of the

forcing and (also) found a transition from a stable/steady to an unsteady/eddy-shedding state for increasing strength and/or decreasing length scale of the forcing, which is not fully captured by a simple reversal of the PV gradient.

The different phenomenon of eastward shedding has emerged more gradually through independent studies focusing on meteorological implications on the one hand and on chemical transport on the other. Postel and Hitchman (1999) noted that there was frequent breaking of Rossby waves over the North Pacific in the upper troposphere in NH summer and suggested that

that the low PV part of the characteristic pattern of wave breaking originated in the Asian anticyclone to the west. Enomoto et al. (2003) studied the dynamics of the Bonin high which is a characteristic feature of the western North Pacific circulation in late NH summer and occurs near Japan. They showed in a modelling study that the anticyclone could result from the roll-up of a low PV filament that had been drawn out of the monsoon anticyclone by fluctuations on the mid-latitude jet, leading to the formation of isolated anticyclones to the (north-)east of the bulk anticyclone. Note, however, that Enomoto et al. (2003)

emphasised the importance of an external Rossby wave source (the 'Silk Road cooling' to the west of the monsoon) in order to form these isolated vortices in their simulations. As we show in Section 3 we can reproduce a similar anticyclonic feature to the north-east of the main monsoon without any extra local wave source.

Later papers such as Garny and Randel (2013) and Vogel et al. (2014) identified the chemical signatures of these dynamical events forming isolated anticyclones via an interaction of the monsoon anticyclone and perturbations on the mid-latitude jet

and referred to them as 'eastward-shedding' events. Note that eastward shedding is rather distinct from a further consequence of time variation of the anticyclone identified by Dethof et al. (1999) where the time variation of the anticyclone and the jet often leads to a filament of low PV extending eastward and poleward into the extratropical stratosphere and subsequently being absorbed into the stratospheric air mass through mixing, perhaps also forming a coherent anticyclonic vortex during this process.

Although different types of shedding events associated with perturbations to the main anticyclone have been identified by monsoon researchers, there is still uncertainty about precise mechanisms. Furthermore the connection between shedding events and other phenomena such as potential bimodality of the monsoon anticyclone Zhang et al. (2002) is not always made clear.



Indeed some of the features that have been previously identified by other authors in their studies of the monsoon anticyclone can, in retrospect, be identified as signatures of eddy shedding. For example, Hoskins and Rodwell (1995) analyse the short-
term response to three-dimensional semi-realistic heating distribution and obtain some agreement between the instantaneous model stream function field and the climatological JJA-averaged stream function obtained from re-analysis data. However they identify a 'major defect' of their (instantaneous) model response in the form of a tendency of the anticyclone to 'split into three separate centres, with additional maxima near east Africa and Japan' that is not clearly visible in the time-averaged re-analysis fields. Enomoto et al. (2003) later identified the the 'defect' anticyclone over Japan as a representation of the Bonin high.

Ren et al. (2015) used a composite analysis and PV budget calculations to investigate mechanisms corresponding to eastward extension-phases of the monsoon anticyclone. They find a strong correlation between the eastward extension and anomalous heat and rainfall patterns over east Asia, indicating a potential meteorological impact of eastward shedding events. Similarly, Luo et al. (2017) describe and analyse an east-west oscillation event of the monsoon anticyclone during 2016. These oscillation events can include a split of the monsoon anticyclone and a zonal shift of its geopotential height maximum. Luo et al. (2017)
further show that this shift can lead to a substantial zonal mass flux and thus might have important implications for horizontal transport. They further link the oscillatory behaviour to a bi-modality of the monsoon, as it was described by Zhang et al. (2002). A unified explanation of the zonal and temporal variation seen in all these phenomenon is as a consequence of eastward and westward shedding events.

### 1.4   Outline

Our aim in this paper is to clarify the mechanisms that control the spatial structure of the monsoon anticyclone, particularly its zonal scale, and its time variability.

In particular we address the following questions:

– Can we find a mechanism that leads to a zonal localisation of the time mean response and does not rely on strong mechanical damping throughout the atmosphere?

– Does the response to a steady localised forcing in a 3D numerical model show evidence for westward eddy shedding similar to the results that have previously been reported for single-layer models with a transition to an eddy-shedding state as the forcing strength increases?

– What influence does the interaction with mid-latitude dynamics have on the locally forced monsoon anticyclone and what role does it play in the phenomenon of eastward eddy shedding?

We address these questions by using a minimal model, based on three-dimensional primitive-equation dynamics, in which the monsoon circulation is forced by an imposed localised heating and the background zonal flow is determined by imposing a meridional (equator-pole) temperature gradient, rather than by being determined (via initialisation or relaxation) by a meridional flow profile that approximates observations. The strength of the localised heating and of the meridional temperature gradient are varied and the implications for the monsoon anticyclone are determined.





The structure of the paper is as follows. In Section 2 we describe the numerical model used in the study and also the re-analysis dataset that provides examples for comparison with the modelled behaviour. Section 3 focuses on the time mean response to localised heating, with particular focus on the zonal scale of the response. In Section 4 we then investigate the different forms of temporal variability of the response and its dependence on the forcing magnitude and the basic state. Finally we give a brief summary of our findings in Section 5.'

## 2    Model and data used in this study

The present study involves numerical experiments with a three-dimensional atmospheric model, as well as various diagnostic analyses of re-analysis data. This Section outlines the specifics of the model and the dataset used.

### 2.1    The numerical model

Numerical experiments are performed using the dry non-linear primitive equation model IGCM1 developed at the University
of Reading (Hoskins and Simmons, 1975). In the model horizontal dimensions are represented via a spherical harmonics series truncation at total wave number 42, corresponding to resolution of about $2.8°$ at the equator. The vertical dimension is represented by 40 $\sigma$-levels (ratio of pressure and surface pressure) equally spaced on the log-pressure scale defined via $z = -H \ln \sigma$, where $H = 7$ km is a scale height. This gives a vertical level spacing of 0.7 km up to the model top at about 28km. Throughout this paper we will use the log-pressure height $z$ rather than $\sigma$-coordinates.

All experiments are run using a Held-Suarez-like (HS) basic state, which is obtained by relaxing the temperature of the system at a rate $\epsilon_r(\phi, \sigma)$ towards the restoration profile $T_r(\phi, \sigma)$, both defined in Equation 1. The corresponding approach was suggested by Held and Suarez (1994) and gives an easy and reliable way to produce a simple representation of the large scale circulation of the mid-latitude atmosphere in the form of a mid-latitude mean zonal jet and a strong temporal variability due to baroclinic instability. In addition to the thermal relaxation the basic state set-up includes a simple representation of surface
friction in the form of a linear damping of horizontal winds at a rate of 1/day at $z = 0$, gradually reducing to zero at about $z = 2.5$ km (exact implementation follows the definition in Held and Suarez (1994)).

$$T_r = \max\left[ T_{strat}, \left( T_{surf} - \Delta T \sin^2 \phi - T_{as} \sin \phi - \Delta\Theta \ln \sigma \cos^2 \phi \right) \sigma^{\frac{2}{7}} \right], \tag{1}$$

$$\epsilon_r = \epsilon_{atmos} + (\epsilon_{surf} - \epsilon_{atmos}) \max\left[ 0, \frac{\sigma - \sigma_b}{1 - \sigma_b} \right] \cos^4 \phi.$$

The parameters appearing in Equation 1 are defined in Table 1. For most of the parameters of the basic state we use the same
values as have originally been used by Held and Suarez (1994), with the exception of the meridional temperature gradient parameter $\Delta T$, which we vary to alter the characteristics (in particular the strength) of the induced mid-latitudinal background flow. Following the proposal by Polvani and Kushner (2002), we further added a term $-T_{as} \sin \phi$ to the restoration profile in Equation 1, allowing us to introduce the hemispherical asymmetry associated with the summer season that is of most relevance to the Asian monsoon. The same approach was adapted by McGraw and Barnes (2016) and Chen and Plumb (2014)





to investigate eddy transport in the lower atmosphere, both choosing $T_{as} = 20$ K, a value twice as large as was originally used by Polvani and Kushner (2002). In this study we use both, a (hemispherically symmetric) 'annual-mean' state ($T_{as} = 0$ K) and an asymmetric 'summer' state ($T_{as} = 20$ K), in order to assess the significance of differences in the structure of the background state.

A tropical monsoon flow is forced by imposing a localised steady heating with structure

$$
\quad Q(\phi, \lambda, z) = \begin{cases} Q_0 V(z) \cos^2\left(\pi \frac{\phi - \phi_0}{2r_0}\right) \cos^2\left(\pi \frac{\lambda - \lambda_0}{2r_0}\right) & \text{if } |\phi - \phi_0| < r_0 \text{ and } |\lambda - \lambda_0| < r_0 \\ 0 & \text{otherwise} , \end{cases} \tag{2}
$$

where $\phi$ and $\lambda$ represent latitude and longitude, respectively, $Q_0$ controls the magnitude of the heating, $r_0$ its horizontal extent and $V(z)$ its vertical structure. The forcing is centred at $\phi_0 = 20°$ latitude and $\lambda_0 = 80°$ longitude, with a radius of

$r_0 = 10°$. Note that the zonal position of the forcing is completely arbitrary and has no influence on the results, since we did not impose any zonally asymmetric features other than the explicit heating. For easier visualisation of the response we defined the range of longitudes in our domain to be $-120° \leq \lambda < 240°$.

The vertical structure $V(z)$ is given by

$$
V(z) = \begin{cases} \sin\left(\frac{\pi}{2} \frac{z}{z_{max}}\right) & \text{if } z < z_{max} \\ \sin^2\left(\frac{\pi}{2} \frac{z_{top} - z}{z_{top} - z_{max}}\right) & \text{if } z_{max} \leq z < z_{top} , \\ 0 & \text{otherwise.} \end{cases} \tag{3}
$$

i.e., the heating is confined to a height below $z_{top}$ and maximises at $z_{max}$. We chose different a structure of $V(z)$ above and below the maximum at $z_{max}$ in order to reduce the vertical gradient of the heating at lower levels, which weakens the corresponding PV forcing and therefore de-emphasise the cyclonic lower level part of the forced monsoon response. However, the results presented in this study do not rely on the precise structure of the forcing.

In each simulation the initial state is that of an isothermal resting atmosphere and the flow is allowed to evolve in the absence

of localised forcing ($Q_0 = 0$) until it reaches what is effectively a statistically steady state. At least 1000 days was allowed for this. The imposed localised forcing (Equation 2) with $Q_0 \neq 0$ is then switched on smoothly over 25 days and subsequently kept steady. Temporal mean states are obtained by averaging the corresponding field for at least 3000 days starting 1000 days after the switch on of the localised forcing. Anomalies are computed as differences between model runs with and without imposed heating. To prevent the build-up of energy at small scales the model includes a parametrisation of sub-grid-scale processes in

the form of an 8th order hyper-diffusion acting on temperature, vorticity and divergence and damping the smallest represented spatial scales on a time scale of 0.1 days.

Table 1 gives an overview of the parameter ranges for the imposed heating and the basic state used in our experiments.





**Table 1.** Physical parameters used in the GCM.

| Symbol | Physical meaning | Value Range |
|:---:|:---:|:---:|
| $Q_0$ | Forcing magnitude | 0.5-10 K/day |
| $z_{max}$ | Height of maximum heating | 10 km |
| $z_{top}$ | Maximum height of heating | 15 km |
| $r_0$ | Length scale of the forcing | $10°$ |
| $\phi_0$ | Meridional forcing centre | $20°$ |
| $\lambda_0$ | Zonal forcing centre | $80°$ |
| $T_{strat}$ | Homogeneous temperature of the stratosphere | 200 K |
| $T_{surf}$ | Surface temperature at the equator | 315 K |
| $\Delta T$ | Meridional temperature gradient parameter | 0-60 K |
| $T_{as}$ | Hemispherical asymmetry parameter | 0K and 20 K |
| $\Delta\Theta$ | Vertical temperature gradient measure | 10 K |
| $\epsilon_{atmos}$ | Homogeneous damping rate of the bulk atmosphere | 0.025 day$^{-1}$ |
| $\epsilon_{surf}$ | Surface damping rate at the equator | 0.25 day$^{-1}$ |
| $\sigma_b$ | Boundary layer depth | 0.7 |

We checked that all results shown are not sensitive to changes in the hyper-diffusion time scale and the length of spin-up or averaging periods. Further, the response of the model in the upper troposphere/lower stratosphere region was not sensitive to
small changes in the details of the bottom drag layer.

## 2.2    Re-analysis data

Part of our analysis is based on the ERA-Interim (ERA-I) re-analysis dataset (Dee et al., 2011) of the European Centre for Medium-Range Weather Forecasts (ECMWF). The data used in this study is a standard product of the re-analysis data set and is given on a horizontal $1°\times1°$-grid following the 370 K isentropic surface. Seasonal means are calculated from monthly
averaged fields for the years 2000-2009, while snapshots show fields at 12:00noon UTC.



## 3 Zonal scale of the time mean response

### 3.1 Localisation due to mid-latitude dynamics

As mentioned earlier the monsoon anticyclone is associated with a pronounced PV minimum in the UTLS region. Figure 1 shows the horizontal structure of this minimum on the 370K isentropic surface, as it appears in the mean over several NH summers (JJA for 2000-2009). Note that only a limited range of longitudes is included in the figure.

The region of low PV ($\leq 2$ PVU) has meridional and zonal widths respectively about 2000km and 10000km. As explained in Section 1 it remains an open question as to what physical or dynamical processes control the zonal localisation of the anticyclone. In this section we conduct a series of numerical experiments to study this problem. In particular we investigate the changes in the response to monsoon heating when varying the meridional temperature gradient parameter $\Delta T$ and the asymmetry parameter $T_{as}$ (see Equation 1) and thus change the properties of the basic state, including the strength, position and structure of the background mid-latitude jet and the corresponding baroclinic eddies.

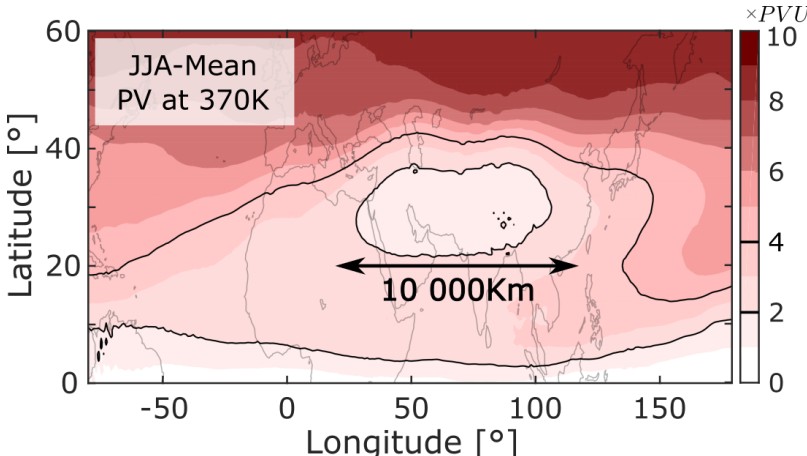

**Figure 1.** Years 2000-2009 JJA-average of ERA-I potential vorticity on the 370 K isentropic surface. The 2 and 4 PVU contours are emphasised and a length of 10 000 km at $20°$ latitude is indicated.

We use the numerical model configuration described in Subsection 2.1. Figure 2 shows the time mean PV structure and the corresponding anomaly from the basic state on the 340 K isentropic surface for a forcing with magnitude $Q_0 = 5$ K/day and a resting background atmosphere, i.e. $\Delta T = 0$ and $T_{as} = 0$. The chosen isentropic level roughly corresponds to $z = 13$ km at the latitudes of the heating, and hence lies between the height of maximum heating ($z_{max} = 10$ km) and the top of the heating ($z_{top} = 15$ km) and in a region with strong negative PV forcing due to the large negative vertical gradient of the imposed heating profile. We can therefore expect to observe a relatively strong anticyclonic PV response on this level. From Figure 2 it is apparent that the PV response is zonally elongated and not confined to the vicinity of the heating. Zonally elongated structures are visible in both, the full PV field and the anomaly field (recall that the zonal position of our forcing is arbitrary due to the zonally symmetric basic state, but for ease of comparison we have centred the forcing at $80°N$).



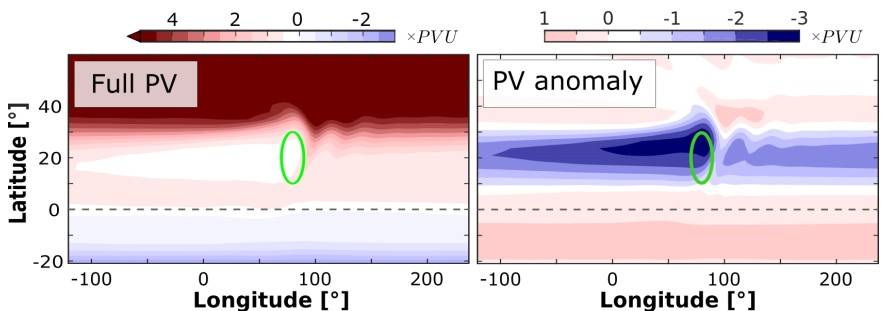

**Figure 2.** Time-averaged full PV response (left) and PV anomaly (right) on 340K for a resting basic state with $\Delta T, T_{as} = 0$ to a forcing with $Q_0 = 5$ K/day. Ellipses indicate the horizontal extent of the heating.

Unlike the Gill-Matsuno model (see Section 1), our model does not include any mechanical friction above the boundary layer. As a result, the forced Rossby-wave response in the upper troposphere decays only slowly with distance to the west of the forcing, with the controlling process being the weak thermal damping in the bulk atmosphere (time scale of about 40 days)[1]. This zonal non-localisation of the monsoon anticyclone is in strong contrast to what is found in re-analysis date (e.g.,

Figure 1) and indicates that additional physical or dynamical processes have to be included in the model in order to obtain a monsoon anticyclone response with a realistic zonal scale.

As a next step we modify the basic state of our system by choosing $\Delta T = 60$ K in Equation 1 and thus making the basic state baroclinically unstable at mid-latitudes (for now we keep $T_{as} = 0$). The resulting dynamics leads to the formation of mid-latitude jet streams with associated highly variable baroclinic eddies.

Figure 3 shows the horizontal structure of the time mean PV and stream function ($\psi$) response on the 335K isentropic surface and the 13 km height plane, respectively[2]. The streamfunction $\psi$ represents the non-divergent part of the horizontal flow. Since we are considering relatively long time scales on which the flow is close to geostrophic balance, the streamfunction is expected to be a useful representation of the actual horizontal flow, which will be non-divergent to good approximation.

The top panel of Figure 3 shows the full anomaly response to a localised steady heating in a baroclinically unstable atmo-

sphere. Two main features can be seen:

- a zonally confined patch around the forcing region (green circle) and

- an elongated zonal band of anomaly to north of the forcing.

The bottom panel of Figure 3 only shows the zonally asymmetric (azonal) component of the $\psi$ and PV response, i.e., with zonal mean subtracted. We find that the anticyclone response in form of a confined anomaly patch remains mostly unchanged

as we remove the zonal mean, while the northern, elongated feature disappears almost entirely. These results suggest that the

---

[1] A damping time scale of 40 days (as specified in our model for the upper troposphere, see Section 2) and a propagation speed of the response away from the forcing region with a velocity of 10 m/s (which is consistent with our findings in Section 4) would lead to an extent of the response of about 35000km (consistent with figure 2).

[2] Note that the height of maximum negative PV forcing corresponds to slightly different isentropic surfaces for different values of $\Delta T$.

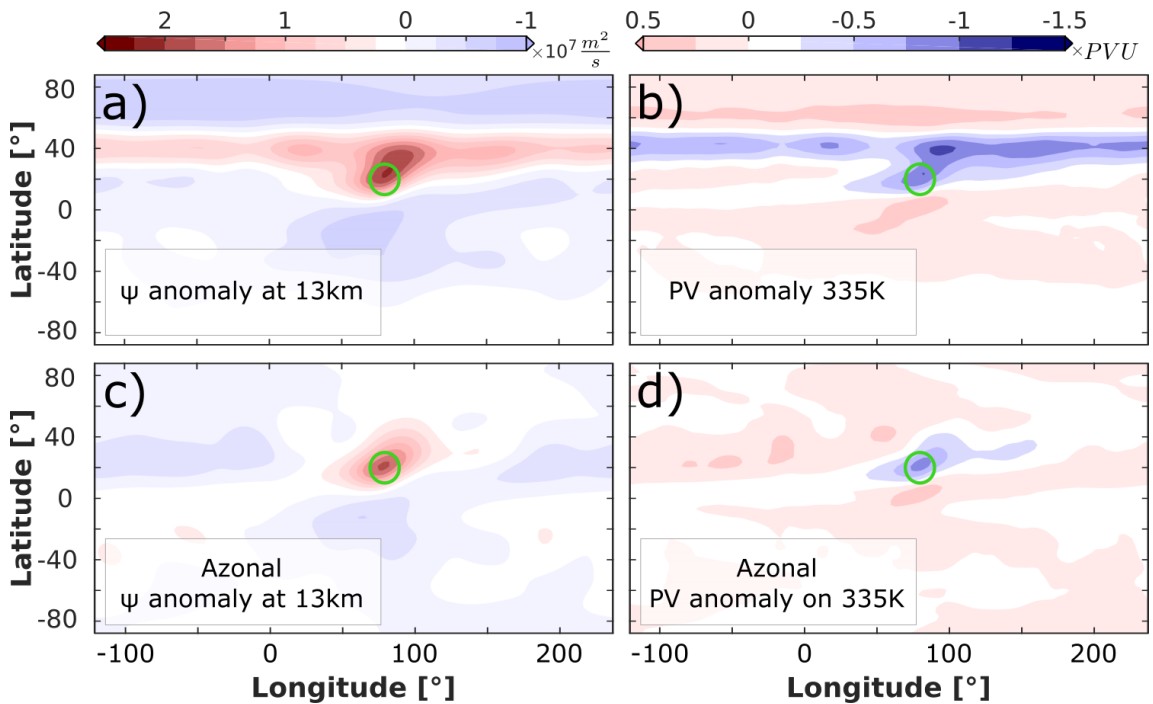

**Figure 3.** Time mean stream function anomaly on the 13km surface (left) and potential vorticity anomaly on the 335K isentrope (right) for an experiment with baroclinically unstable background, i.e. the symmetric HS state with $\Delta T = 60$ K. The forcing magnitude is $Q_0 = 5$ K/day. Bottom panel shows plots with zonal mean removed (azonal anomaly). Circles indicate position and extent of the forcing region.

change to the background circulation as associated with non-zero $\Delta T$ has resulted in a subtropical monsoon anticyclone that is much more zonally confined that was found with $\Delta T = 0$, but the role of the zonally elongated response at mid-latitudes cannot immediately be dismissed and we therefore examine this further in the following subsection.

## 3.2 Distinction between response to zonal mean and zonally varying part of the forcing

To further study the nature and relative importance of the zonal band structure seen in Figure 4 we conduct an experiment using an imposed forcing identical to the one described Section 2, but without a zonal mean component (the zonal mean of the original heating was subtracted along the entire latitude band). Figure 4c shows the response of the system to the forcing without zonal mean component. As can be seen, the zonally symmetric band-structure is not present in this case and the PV anomaly response looks almost identical to the response in the case with full heating (including zonal mean), when subtracting

the zonal mean of the final response (Figure 4b). This indicates that the response to the zonal mean part and to the azonal part of the heating are almost independent and the zonally extended feature is not caused by the 'local contribution' of the forcing but only by its zonal mean component.



In particular the localisation to the west of the forcing is similar in Figures 4b and c, despite the corresponding basic states differing slightly in their zonal wind structure (due to the changes in wind corresponding to the zonal band structure, also seen

later in Figure 5). This suggests that the direct advection by the mean flow is less important in localising the response and thus that the stirring effect of the baroclinic eddies (which is the other physical ingredient included as a result of non-zero $\Delta T$) potentially plays a significant role. Note that Figures 4b and c also show indication of an (north-)eastward extension of the time-mean response of the forced anticyclone. We will come back to this feature in Section 4 and link it to the phenomenon of eastward eddy shedding.

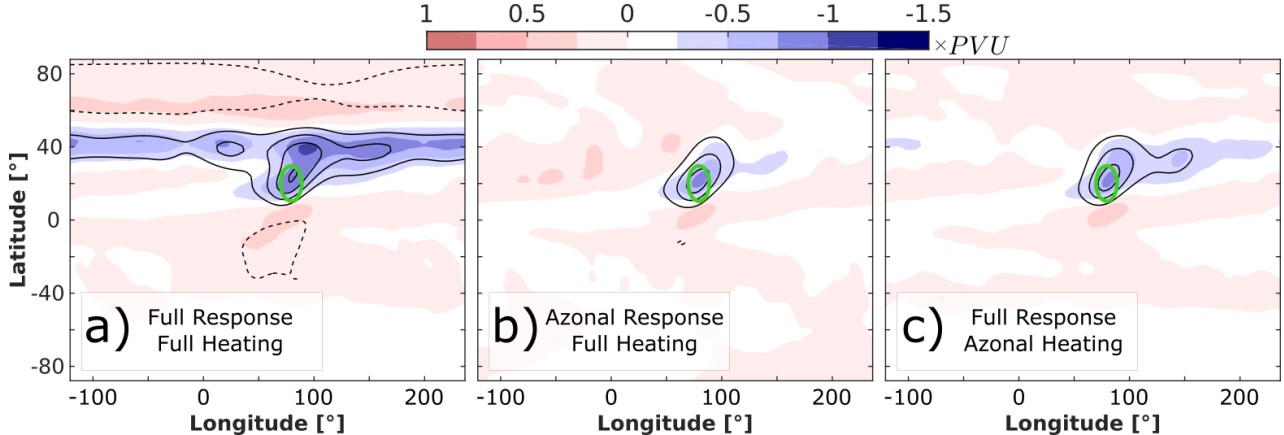

**Figure 4.** Mean PV anomaly at 335 K (shading) and mean $\psi$ anomaly at 13 km (contours) for the symmetric HS basic state with $\Delta T = 60$ K. Contour interval is 5 km$^2$/s. Subplot a) shows the full anomaly response to the full heating. Subplots b) and c) show the azonal response to the full heating and full anomaly response to a heating without zonal mean, respectively.

The fact that the zonally extended band-structure is caused by the zonal mean component of the heating suggests it to represent an annular-mode-like response of the system. Annular modes are hemispheric scale patterns of climate variability and, in terms of empirical orthogonal functions (EOFs), the leading mode of mid-latitudinal variability (see, e.g., Thompson and Wallace (2000) for a detailed description on annular modes and examples for the observed variability of the atmosphere). A perturbation of the global temperature structure and flow fields by an imposed forcing can cause a response similar to such

an annular mode. Effectively, the structure and/or position of the resulting baroclinic jet and eddy features can change as a result of a perturbation. This might lead to a meridional shift of the time mean jet and can thus manifest as a zonally symmetric PV or $\psi$ structure.

    Various studies (e.g., Ring and Plumb, 2007; Butler et al., 2010) have shown that simple dry GCMs with basic states similar to the HS configuration can indeed exhibit an annular-mode-like response when perturbed by a zonally symmetric forcing. As

we have shown in Figure 4, the response to the zonally symmetric part of the heating is manifested in the zonally symmetric part of the response, which can be separated from the zonally localised monsoon anticyclone response forced by the azonal part of the heating.



A characteristic of annular-mode-like responses, in contrast to a simple advection of low (monsoon) PV by the mean flow, is that the coupled system of mean winds and the baroclinic eddies responds as a whole. Figure 5a shows the time and zonal mean

zonal wind profile of the HS basic state (contours) and the mean eddy momentum transport, given by the temporally averaged meridional eddy advection of zonal wind $\overline{u'v'}$ (shading), where $u'$ and $v'$ describe the deviation of zonal and meridional wind from the zonal mean, respectively, and the overbar indicates the zonal mean. The mean eddy momentum transport is a measure for the strength of the baroclinic eddies and indicates their role in driving the mean flow of the system. Figure 5b shows the corresponding modification of the zonal wind and eddy flux when the system is forced by a local heating with $Q_0 = 5$ K/day.

A polewards shift of the jet can be seen, driven by changes in $\overline{u'v'}$. Note that the mean flow and the eddies seem to respond as a coupled system and the response is consistent with the typical EOF response associated with annular modes (e.g., Butler et al. (2010)).

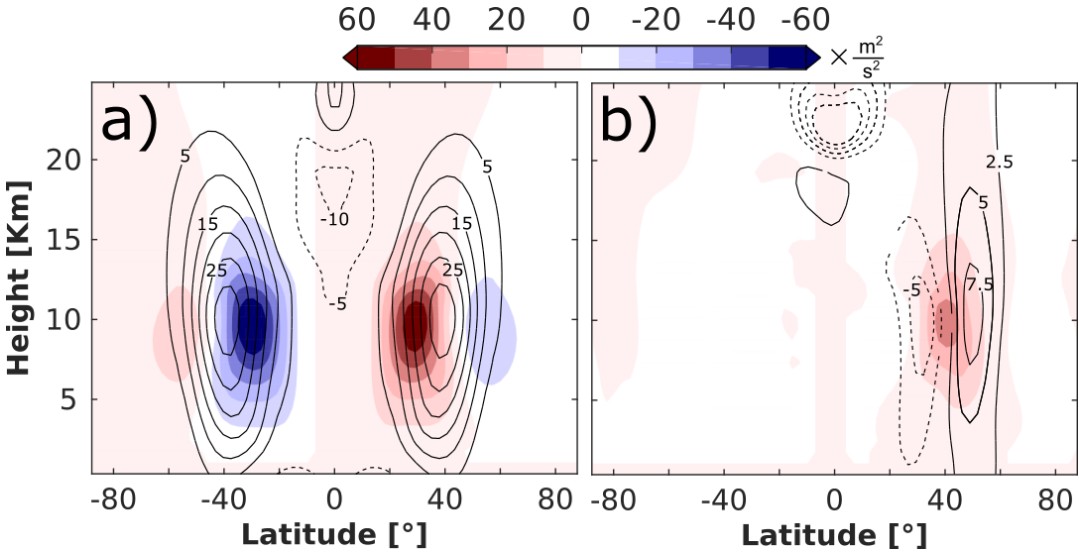

**Figure 5.** Subplot a) shows time and zonal mean zonal wind field of the symmetric HS basic state (contours) with $\Delta T = 60$ K, and the time and zonal mean eddy momentum flux $\overline{u'v'}$ (shading). Subplot b) as a), but anomalies from the basic state caused by a forcing with magnitude $Q_0 = 5$ K/day.

The results presented above suggest that the elongated mid-latitude feature shown in Figure 3 has no particular significance with regard to the monsoon anticyclone itself and hence that the addition of the meridional temperature gradient and the

resulting baroclinic instability and jet formation is the key ingredient which results in zonal localisation of the monsoon anticyclone response to the confined heating. One has to be careful when interpreting Figure 5 since the shown anomalies include the localised anticyclone structure in the forcing region that can be seen in Figure 5b. However, since the change in mean zonal wind is centred at about 40° latitude, and hence north of the forcing, it seems to indeed mostly capture the annular-mode-like feature.





### 3.3 Sensitivity of the response to details of the meridional temperature gradient


We now investigate in more detail how the monsoon anticyclone response to a confined steady heating changes as we gradually vary the meridional temperature gradient parameter $\Delta T$. This will lead to a change in zonal jet strength, centre position and shape, as well as corresponding changes in the baroclinic eddies. Figure 5 shows the meridional profiles of the mean zonal wind at 13 km. It can be seen how the jets generally shift polewards and become stronger as $\Delta T$ is increased. Note that the

zonal wind and meridional shear within the forcing region ($10 - 30°$ latitude) is almost the same for all shown values of $\Delta T$ (the exception is the case with $T_{as} = 20$ K which we will discuss in a later section).

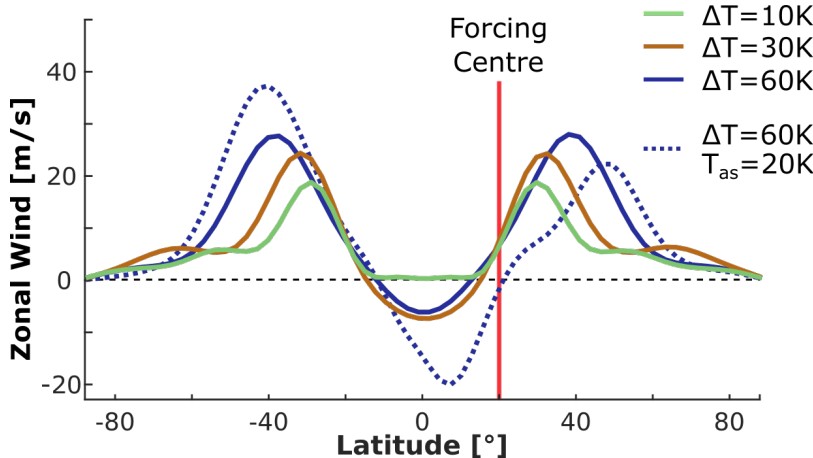

**Figure 6.** Time and zonal mean profiles of zonal wind at 13km for basic state profiles with different values of the meridional temperature gradient parameter $\Delta T$. The blue dotted line shows the summer state with $T_{as} = 20$ K, all other profiles show the annual-mean HS state with $T_{as} = 0$ K. The vertical line indicates the centre of the forcing region at $20°$ latitude.

Several studies have previously investigated the effect of varying physical parameters in HS-like states, e.g., Zurita-Gotor (2008) who found a similar tendency for poleward-shift, strengthening and broadening of the jets when changing the meridional temperature gradient.

The altered background state does affect the time mean response of the flow to the localised heating, as can be seen in Figure 7. As we increase the meridional temperature gradient $\Delta T$ we see how the distinction between the two features discussed in Section 3.2 (localised anticyclone and zonal-band-structure) becomes more apparent. First, the response gradually becomes zonally localised at the latitudes of the heating, forming a coherent anticyclone in the forcing region. Second, the zonal mean component of the anomaly response seems to shift northward, which can be interpreted as formation of an annular-mode-like response, as shown earlier. Figure 5 shows the jet maximum tends to move polewards as $\Delta T$ increases and a similar polewards

shift of the corresponding EOF would not be surprising.

In the rest of this subsection we investigate the response to a steady confined heating for a summer basic state in which hemispheric asymmetry has been incorporated by setting $T_{as} = 20$ K, as discussed in Section 2. The intention is not to contrast



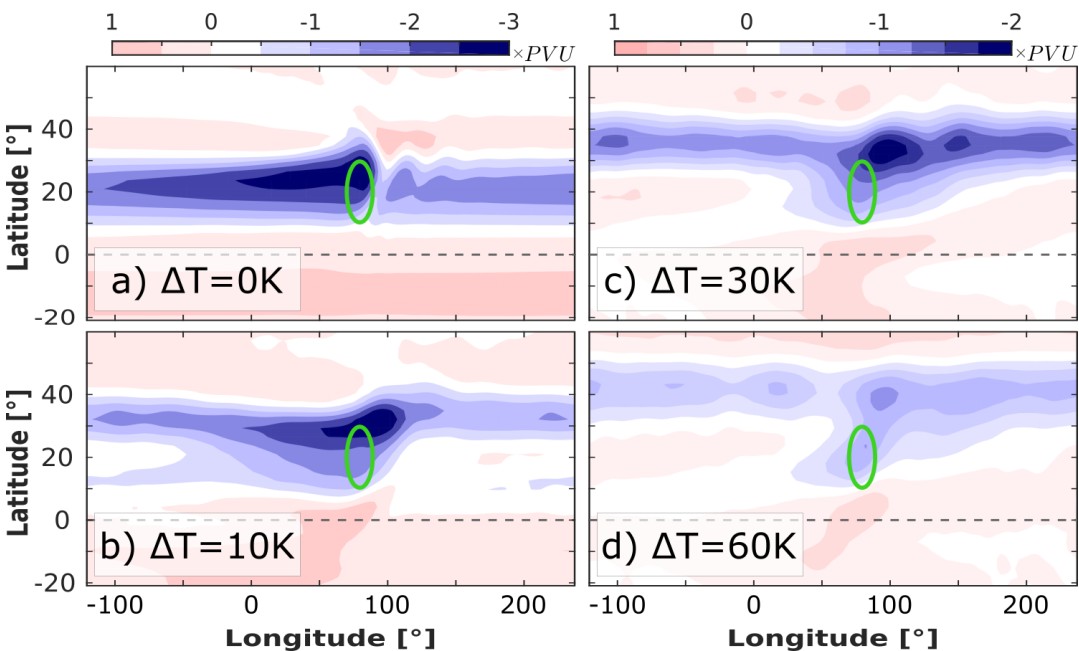

**Figure 7.** Time mean PV anomaly response at 340 K (left panel) and 335 K (right panel) for a forcing with magnitude $Q_0 = 5$ K/day and baroclinic HS basic states with $T_{as} = 0$ and different values of meridional temperature gradient parameter $\Delta T$. Ellipses indicate the forcing region.

the hemispheres, but to obtain a basic state that is within the northern hemisphere (where the forcing is a located) a more

realistic representation of the summer state, rather than the more annual-mean-like state of the original HS configuration with $T_{as} = 0$. Figure 8 illustrates the main difference between the basic state mid-latitude jet profiles for the annual-mean and summer states. Note in particular the zonal wind for the summer case at a height of 13 km (Figure 8c) is close to zero in the centre of the forcing region at 20° latitude.

The upper level $\psi$ and PV response of the summer state to a steady localised forcing are shown in Figure 9. As for the

symmetric case (Figure 3), we find an annular-mode response, in the form of a zonally symmetric band to the north of the forcing region, and a pronounced localised anticyclone confined to the vicinity of the forcing region. The annular-mode-part of the response seems to be broader and span a larger meridional range than that observed in the case with symmetric basic state. The structure of the localised anticyclone, however, seems to be qualitatively similar to the symmetric case. The fact that the localisation of the response in the annual-mean and summer cases are similar, while the zonal winds essentially vanish within

the forcing region (see Figure 8) is further evidence that stirring by baroclinic eddies plays a significant role in localising the response and it is not solely due to the direct advection by the zonal mean flow (also briefly discussed in Section 3.2).

The azonal mean response seems to show zonal extensions, with the response stretching eastward to the north and westward to the south of the main response, giving the response a a North-East/South-West tilted appearance. Similar extensions can be seen in the symmetric HS case (Figure 3) although they do not seem to extend as far out of the forcing region and are weaker

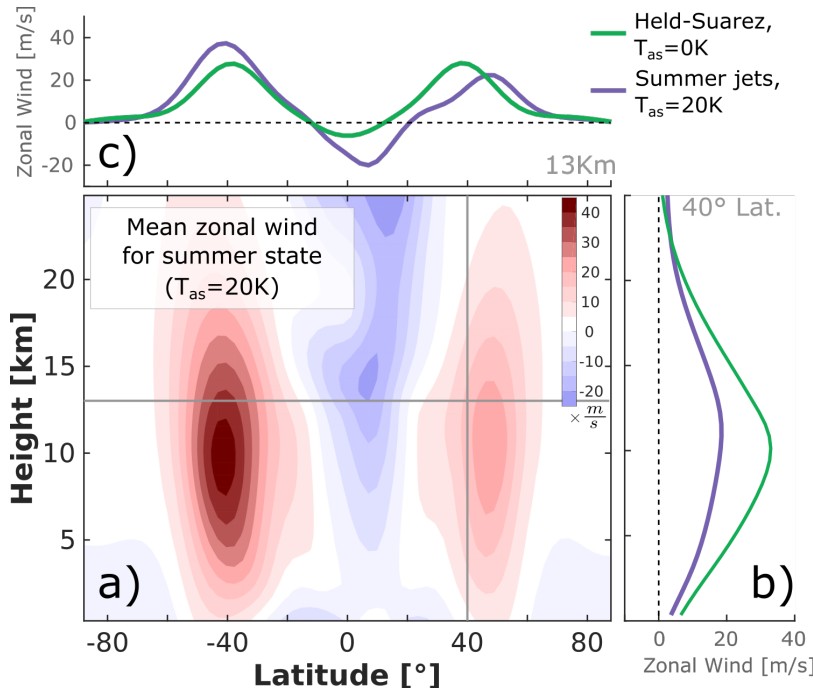

**Figure 8.** Subplot a) shows the zonal and time mean zonal wind profile for the summer basic state, using $T_{as} = 20$ K and $\Delta T = 60$ K. Subplots b) and c) show the zonal jet profiles for the summer and annual-mean cases at 13 km and $40°$ latitude, respectively, indicated by grey straight lines in subplot a).

in magnitude. Further examination (see Section 4) suggests that these zonal extensions arise from east- and westward eddy shedding. As we explain eastward shedding is caused in part by advection by the mid-latitude jet and hence happens preferably closer to the centre of the jet (located north of the forcing), while westward shedding can only occur in the absence of strong eastward winds and hence occurs further to the south (where the zonal winds are weak or even westwards).

In the next section we investigate the time dependence of the monsoon response in a baroclinic background, with special interest in the phenomena of east- and westward shedding.

## 4   Temporal variability in the form of eddy shedding

### 4.1   Eddy shedding in re-analysis data

As noted in Section 1.3, various authors have discussed in detail examples of the evolution of the horizontal monsoon anticyclone PV structure during specific east- or westwards shedding events (e.g., Hsu and Plumb, 2000; Garny and Randel, 2013; Vogel et al., 2014). Figure 10 provides further illustration of shedding events, in this case as observed in re-analysis data during July and August 2000. A localised anticyclone in form of a PV minimum centred at about $30°$ latitude and $70°$ longitude can

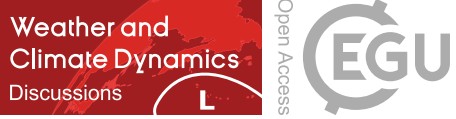

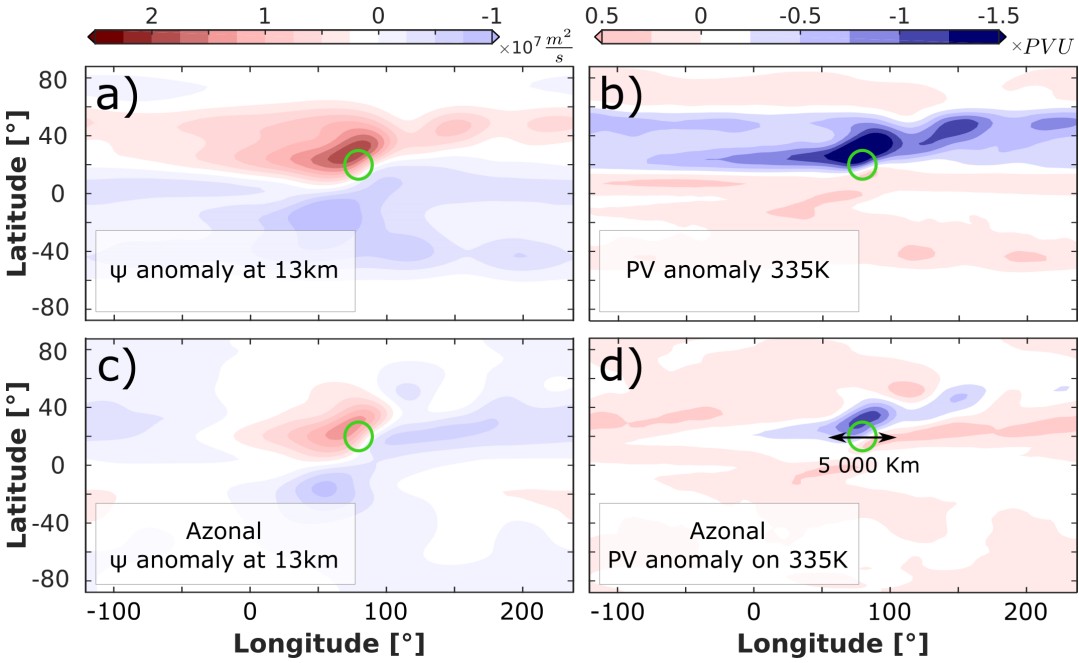

**Figure 9.** Mean stream function (left) and potential vorticity (right) anomaly response to a forcing with $Q_0 = 5$ K/day perturbing the summer basic state with $T_{as} = 20$ K and $\Delta T = 60$ K. The bottom panel shows the zonally asymmetric part of the response. Circles indicate the forcing region and the black arrow indicates a distance of $5\,000$ km at $20°$ latitude.

be identified on 11th of July in Figure 10a. Over the following days the northern edge of the anticyclone becomes distorted and the patch of low PV eventually breaks into two almost equally sized anticyclones on July 19th (Figure Figure 10c). The events between July 11th and 19th are representative for a westward eddy shedding event. In the case shown the broken-off

anticyclone does not propagate westwards very far, but stays rather in place. The reason might potentially be the interaction with the mid-latitude jet and a pronounced baroclinic eddy, which can be seen at about -30° longitude and 40° latitude on 19th of July in Figure 10c . The meridional wind associated with the eddy further seems to disintegrate the shed vortex over the period until August 1st in Figure 10f.

A few days after the westward shedding event, on July 30th in Figure 10e, we can observe an example of an eastward

shedding event. At about 130° longitude one can see how a filament of low PV gets pulled out of the main anticyclone due to meridional advection of a passing by baroclinic eddy. The filament then breaks off, rolls-up and subsequently gets advected eastwards by the mid-latitude jet on August 1st (Figure 10f). Note that, whilst there is significant variation between the daily PV fields shown in the different panels of Figure 10, in each case a clear monsoon anticyclone, manifested as a coherent low PV structure (albeit sometimes split into two) can be clearly identified.


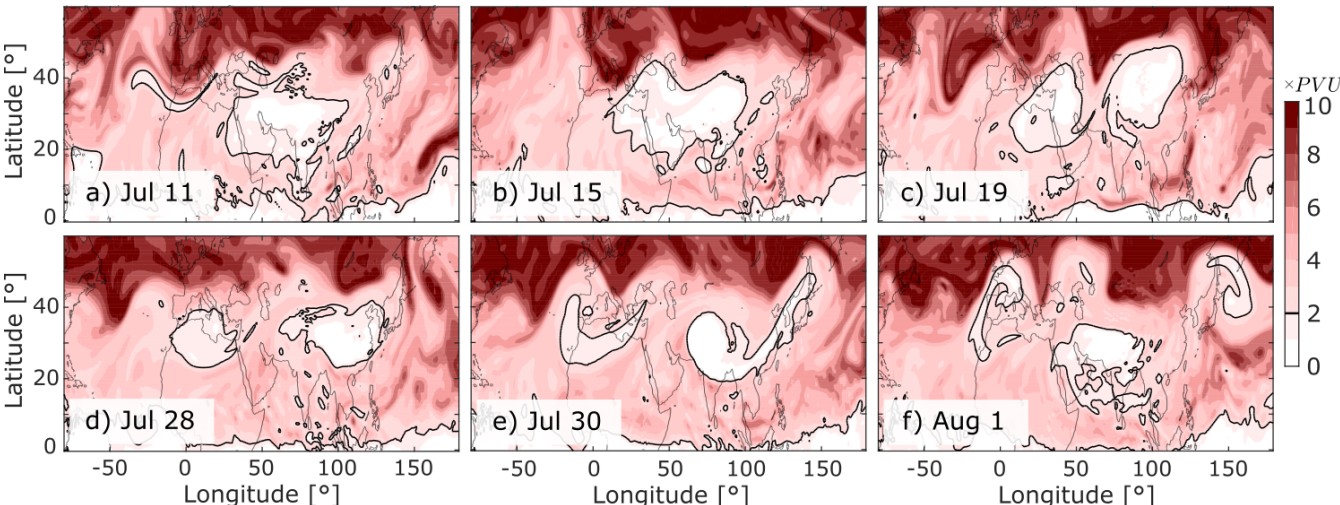

**Figure 10.** ERA-I potential vorticity on the 370K isentropic surface for various days of the year 2000. The 2 PVU contour is emphasised. Subplots b to d display the evolution of a westward shedding event, subplots e and f show an eastward shedding event.

## 4.2 Emergence of westward eddy shedding in model experiments with a basic state at rest

In Section 3 we focused solely on the time-mean monsoon anticyclone response of the flow to a steady localised heating. In this Section we examine the time variability of this response, for a range of forcing and basic state parameters. In particular we show how the phenomena of west- and eastward eddy shedding can spontaneously emerge for certain parameter combinations of the steady forcing and basic state profiles.

As in Section 3 we start by considering the case of the forced response in a resting atmosphere, i.e. choosing $\Delta T = 0$ in Equation 1. The corresponding time-mean response was shown in Figure 2. Figures 11a and b show the instantaneous $\psi$ and PV response to a steady localised heating with two different heating magnitudes, while Figures 11c and d show the corresponding time evolution of the azonal (with zonal mean subtracted) anomaly response along the $20°$ latitude line. A clear qualitative difference between both cases with different forcing strength can be seen. The response to the weak forcing ($Q_0 = 0.5$ K/day) is entirely steady and simply consists of a westwards extending beta-plume which slowly decays with distance to the forcing, probably due to the weak thermal damping of the model (as also mentioned in Section 3.1). In the case with a strong heating ($Q_0 = 5$ K/day) the behaviour of the model is completely different. In this case the system does exhibit a periodic creation and westward shedding of vortices from the region of (steady) heating, inducing a strong temporal and zonal variability. The eddies propagate westward at an essentially constant speed of about 12 m/s, indicated by the constant slope of the diagonal line in Figure 11d. The spontaneously emerging temporal variability of the response to a steady forcing that can be seen in Figure 11 is a potential explanation for the observed phenomenon of westward eddy shedding in relation with the observed monsoon anticyclone (e.g., Figure 10).



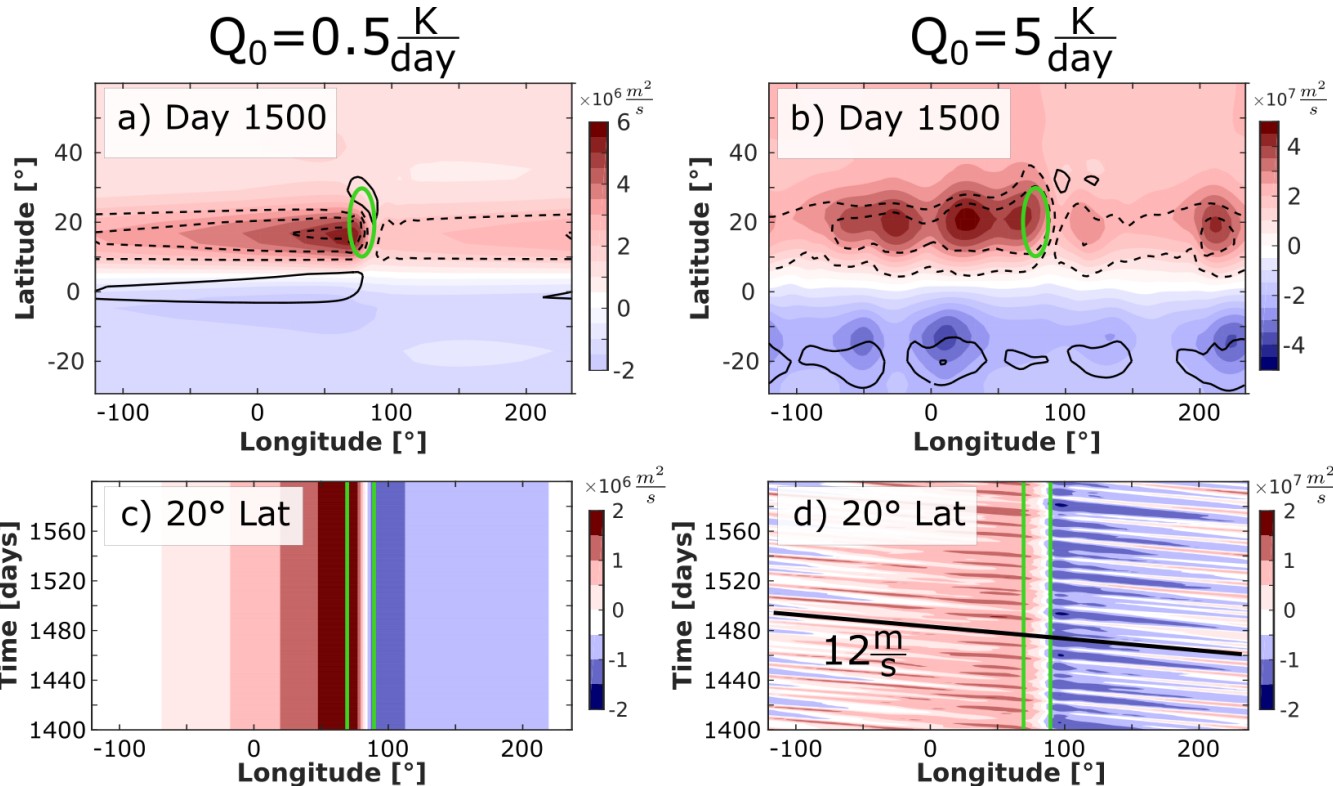

**Figure 11.** Top panel (a+b): Day 1500 snapshot of the stream function anomaly response at 13 km in an atmosphere at rest ($\Delta T = 0, T_{as} = 0$) for two different forcing magnitudes. Contour lines show day 1500 PV anomaly on the 340 K isentropic surface, with dashed contours representing negative values, the zero contour not shown and the contour interval being 0.2 PVU and 1 PVU for subplots a and b, respectively. The ellipses indicate the forcing region. Bottom panel (c+d): Evolution of azonal stream function along line at 13 km and $20°$ latitude for the same two experiments shown in the top panel. Vertical lines indicate the extent of the forcing region, the slope of the diagonal black line in subplot d indicates a reference velocity of 12 m/s.

Note that, as discussed in Section 3, the lack of strong dissipation in our model leads to an extreme westward elongation of the response and an eventual re-emergence of air parcels in the forcing region due to the spherical geometry of the domain.
In cases where the system is in the shedding regime, i.e. with strong heating magnitudes, this phenomenon can, in particular, lead to a 'phase-locking process', where vortices that re-emerge in the forcing region can trigger a new shedding event. Such a phase-locking mechanism can potentially influence flow characteristics like the shedding frequency or the size of shed vortices and the corresponding details of the response have to be interpreted with caution. Other authors have previously investigated eddy shedding in a periodic domain, but either restricted their experiments to early time behaviour (Davey and
Killworth, 1989) or did not mention the phenomenon of phase-locking (Hsu and Plumb, 2000). However, phase-locking is less relevant for experiments in which the response is, via some process, zonally localised, for example experiments that include a representation of mid-latitude dynamics (as discussed in Section 3).



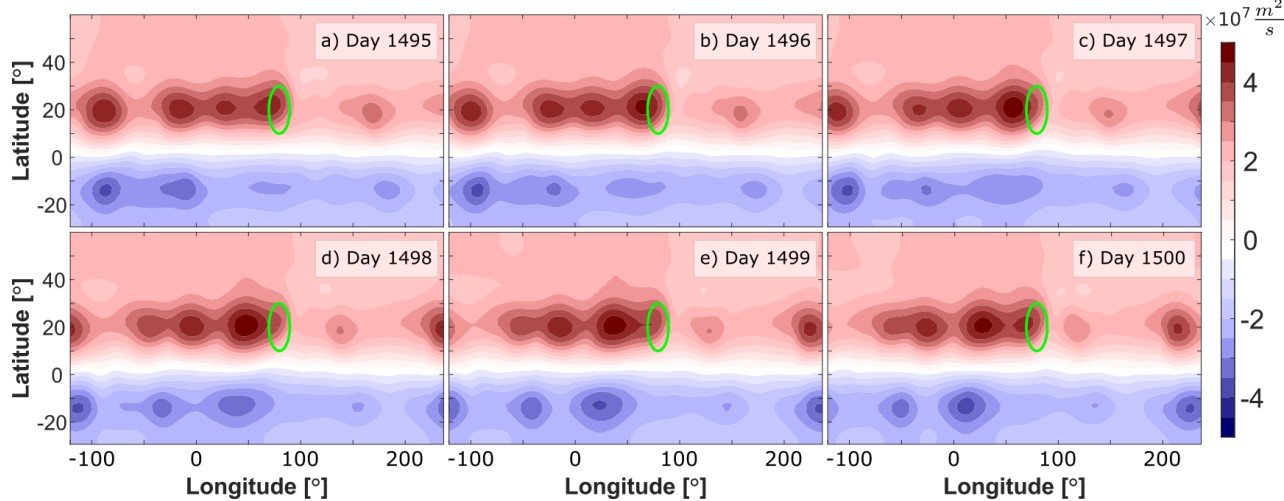

**Figure 12.** Evolution of the stream function response at 13 km over several days in an atmosphere at rest ($\Delta T = 0, T_{as} = 0$) and for a forcing magnitude of $Q_0 = 5$ K/day. The ellipses indicate the forcing region.

In order to obtain a better understanding of how the westward shedding process evolves Figure 12 displays the horizontal distribution of $\psi$ at 13 km for several consecutive days. At day 1495 (i.e., Fig. 12a) we find a localised anticyclone developing

inside the forcing region and slowly strengthening and propagating westward over the next days. As it moves out of and away from the forcing region the stream function inside the forcing region decreases slightly (see Fig. 12d). Once the anticyclone moved sufficiently far away from the forcing the stream function another isolated vortex starts developing inside the forcing region and correspondingly the stream function starts to increase again (Fig. 12f). Figure 12 suggests the shedding process to be characterised by the periodic production of individual vortices inside the forcing region along with a continuous westward

propagation of the strengthening vortex, resulting in alternating periods of high and low equatorward flow anomalies within the forcing region and therefore variations of the southward advection of high background PV. We find the evolution of the stream function response during a westward shedding event in our three-dimensional model to be similar to what has been described by other authors in relation to single-layer studies on westward shedding (Hsu and Plumb, 2000; Davey and Killworth, 1989).

For the rest of this section we focus on experiments in the shedding regime, i.e., experiments with imposed heating of

magnitude and investigate the details of the intrinsically emerging temporal variability for various basic state configurations.

### 4.3 Transition to eastward eddy shedding via introduction of mid-latitude dynamics

Next we investigate how the (transient) behaviour of the response changes as we make the basic state baroclinically unstable by introducing a meridional temperature gradient, i.e., by increasing $\Delta T$ in Equation 2. As mentioned earlier, the baroclinically unstable basic state develops westerly jets and baroclinic eddies in mid-latitudes. The latter induce a strong spatial and temporal

variability to the background flow and hence the PV and stream function fields generally show a range of quickly evolving spatial structures which can correspond to anomalies of fairly large magnitude compared to the time mean field. Since the



response magnitude of the explicitly forced anticyclone strongly depends on the forcing strength (as suggested by Figure 11) one can imagine situations where the response to a steady localised heating is weak relative to the varying anomalies of the background and thus it is difficult to identify a clear and pronounced anticyclone on a day-to-day basis. In situations with
sufficiently strong forcing, and therefore strong response, one would, on the other hand, expect to see a well-defined anticyclone in terms of a coherent low-PV structure (as is typically the case in re-analysis date; see Figure 10).

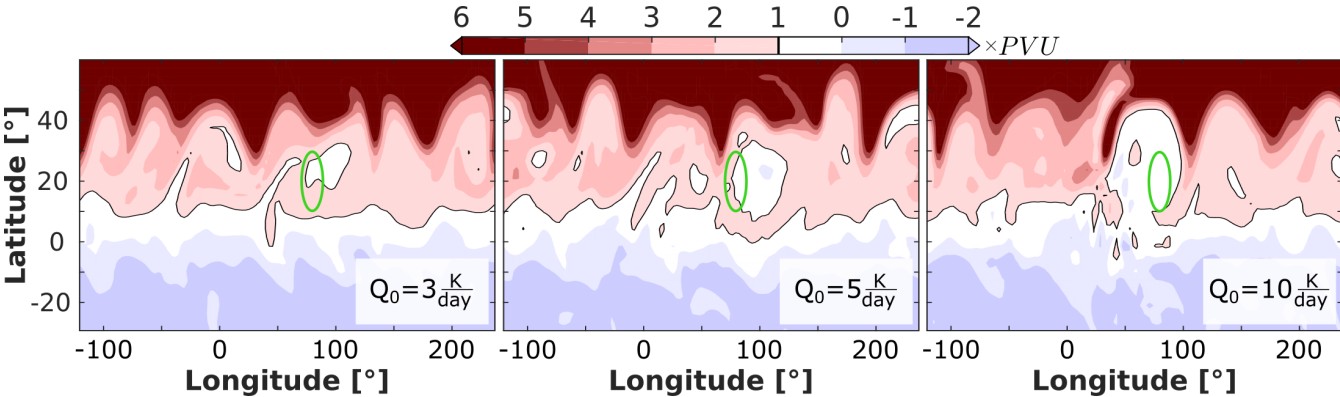

**Figure 13.** Day 1525 PV response at 335 K using different forcing magnitudes and a symmetric HS state with $\Delta T = 60$ K. Ellipses indicate the forcing region and the 1 PVU contour is highlighted.

Figure 13 shows instantaneous snapshots of the PV response to heating distributions with varying magnitude $Q_0$. For the case with $Q_0 = 10$ K/day a clear patch of low PV can be identified in the vicinity of the forcing region, relating to a clearly visible anticyclone, i.e. similar to the observed cases shown in Figure 10. For the weaker heating with $Q_0 = 3$ K/day, on the
other hand, the anticyclone is barely visible and could easily be misinterpreted for a feature of the basic state (e.g., a baroclinic eddy). This dependence of the response amplitude on the forcing magnitude is a persistent and usual characteristic of the flow. Whilst the PV field of course varies on a day-to-day basis, these overall characteristics of the PV field (as the forcing magnitude varies) are robust and reproducible. In all three experiments shown in Figure 13 the PV anomaly of the anticyclone becomes clearly visible when taking long time averages since the azonal anomalies of the background state average out and the monsoon
response creates a uniquely identifiable azonal feature in or near the heating region.

The nature of time dependence of the response changes fundamentally as we gradually increase $\Delta T$ and thus gradually introduce active mid-latitude dynamics. Figure 14 shows the time evolution along latitude lines at the northern edge (top panel) and through the centre (bottom panel) of the forcing region for a range of values of $\Delta T$. In the case where the basic state is at rest (Fig. 14a+b) we find pronounced westward shedding of vortices from the forcing region. The vortices subsequently
propagate westward along the 20° latitude line, as described earlier. The shed eddies do not have a strong PV signature at 30°. In the case with strong mid-latitude dynamics (Fig. 14g+h), on the other hand, almost no evidence for westward shedding can be seen, but we find clear indication of eastward propagating coherent features. These eastward shed eddies have pronounced





PV signatures at $30°$, i.e., to the north of the forcing region and closer to the centre of the mid-latitude jet (see Figure 6), suggesting them to be advected eastwards by the background wind.

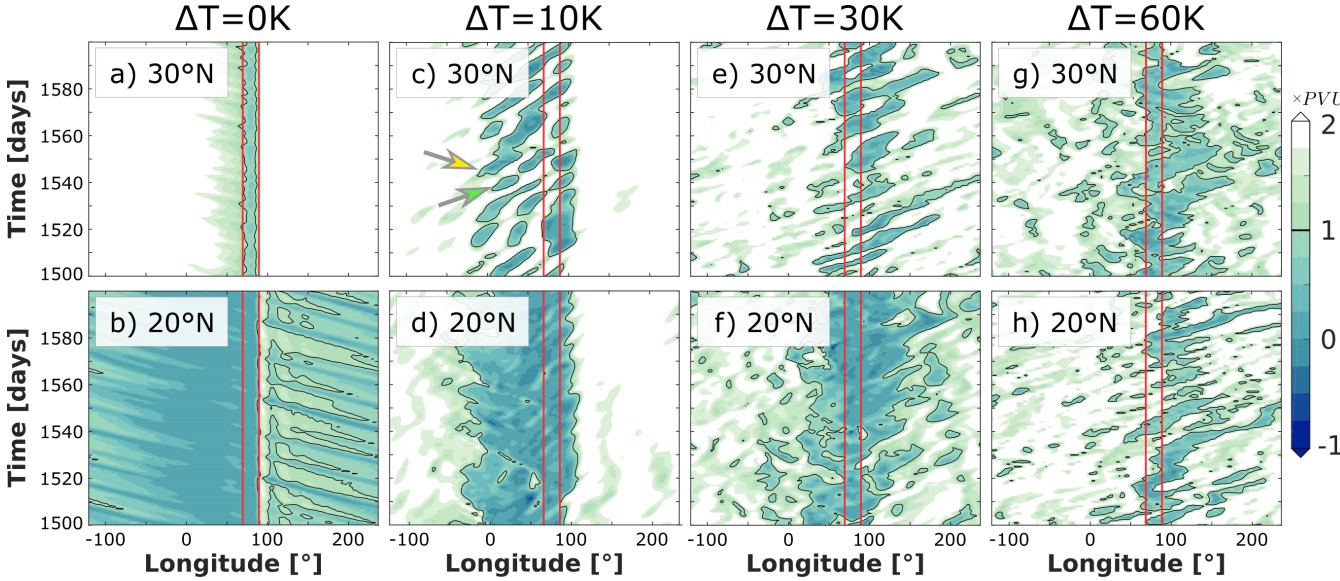

**Figure 14.** Time evolution of PV along two latitude lines on the the 340 K (a-d) and 335 K (e-h) isentropic surface. The forcing magnitude is $Q_0 = 5$ K/day and the basic state uses $T_{as} = 0$ and different values of $\Delta T$. Vertical lines indicate the extent of the forcing region and the 1 PVU contour is highlighted. Arrows show the structures shown in Figure 16.

To gain a better understanding of the dynamical processes leading to the modified temporal variability of the response in experiments with strong mid-latitude dynamics (e.g. Figure 14d+h) it is useful to study in detail the evolution of a specific eastward shedding event. In order to see a clear evolution of the anticyclone on a day-to-day basis (as discussed earlier; see, e.g., Figure 13) we choose a forcing magnitude of $Q_0 = 10$ K/day for this experiment, although we see similar behaviour for weaker magnitudes (as also suggested by Figure 14 for $Q_0 = 5$ K/day).

Figure 15 shows the evolution of PV on the 335 K isentrope over the course of 10 days. Strong wave-like perturbations can be seen along the steep meridional background PV gradient at about $40°$ latitude, associated with the eastwards moving baroclinic eddies. One can further clearly see a pronounced monsoon anticyclone in the form of a PV minimum in the vicinity of the forcing region (green ellipse), almost elliptic in shape at day 1525. At day 1529 a filament of low PV is pulled out of the bulk-PV region by the strong winds of a baroclinic eddy associated with a sharp PV gradient. The filament then rolls up and





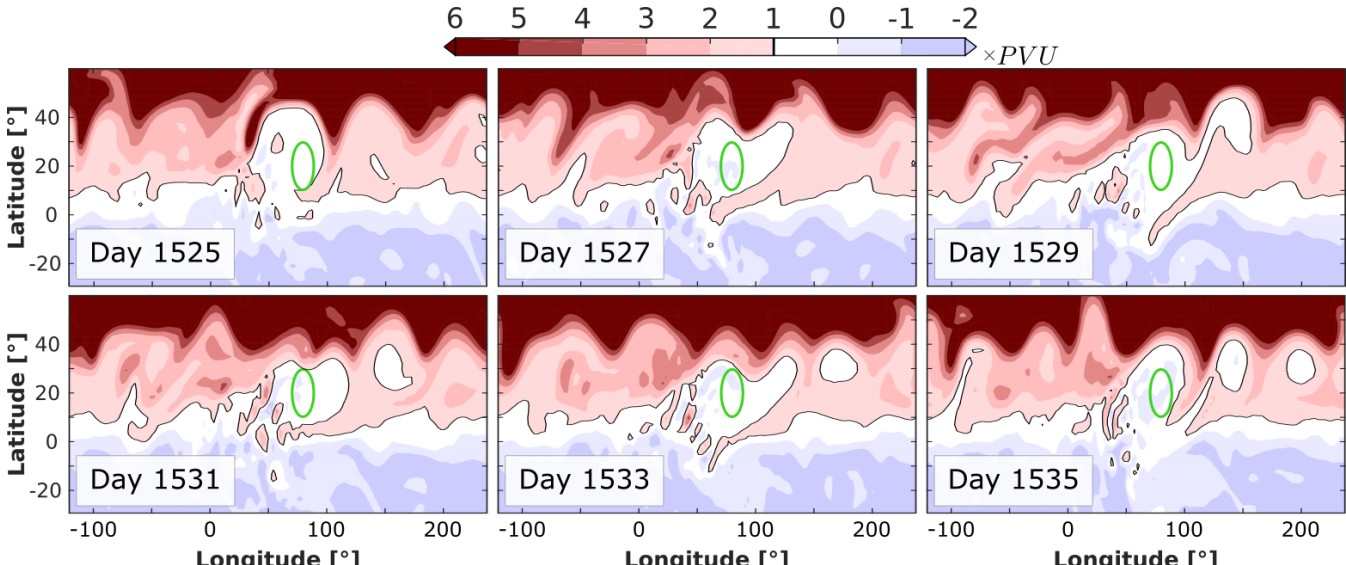

**Figure 15.** PV response at 335 K for various days. The forcing magnitude is $Q_0 = 10$ K/day and we used a symmetric HS state with $\Delta T = 60$ K. Ellipses indicate the forcing region and the 1 PVU contour is highlighted.

breaks off over the following days, forming an isolated anticyclone at day 1531 which subsequently gets advected eastward by the mean zonal wind of the background state. A second such eastward shedding event is happening between days 1531 and 1535. This behaviour, which in our simulations occurs repeatedly as a response to a combinations of steady monsoon heating and statistically steady baroclinic eddy dynamics in the extratropics, is very similar to what has been reported during specific events (e.g., Enomoto et al., 2003; Garny and Randel, 2013). Note that the centre location of eastward shed vortices in Figure

15 is located northward of the forcing region, in agreement with the PV signals observed in Figures 14e and g.

     A slightly different kind of behaviour seems to be happening for the weakly baroclinic background state with $\Delta T = 10$ K. As can be seen in Figure 14d a coherent bulk anticyclone has formed at 20° latitude and between about -20° and 100° longitude. There is no pronounced time variation or preferred direction of propagation obvious and the response seems relatively steady. However, to the north of the forcing region (Figure 14c) patches of anomaly seem to appear to the west of the forcing region,

propagate eastwards towards it and eventually disappear near the eastern edge of the forcing. In the following paragraphs we discuss this state in more detail, before moving on to a (hemispherically asymmetric) summer basic state.

     Figure 16 shows how the PV field on the 340 K isentropic surface evolves between days 1542 and 1550 for a background state with $\Delta T = 10$ K. The arrows highlight the position of two pronounced baroclinic eddies forming a wave-like structure on the mid-latitude PV gradient of the background. As the eddies move eastward, they start to interact with the PV minimum of the

anticyclone. The meridional winds associated with the horizontal gradients in PV pull low PV out of the anticyclone northward on one side and advect heigh-value PV southward on the other. A potential dynamic interaction of the monsoon anticyclone could then further intensify this baroclinic wave structure. It is also possible that the combination of strong westerlies of the





mid-latitude jet and the northern part of the anticyclone form a sort of wave-guide, as pointed out by Enomoto et al. (2003), which additionally favours certain waves and hence leads to a stronger modulation of the PV field.

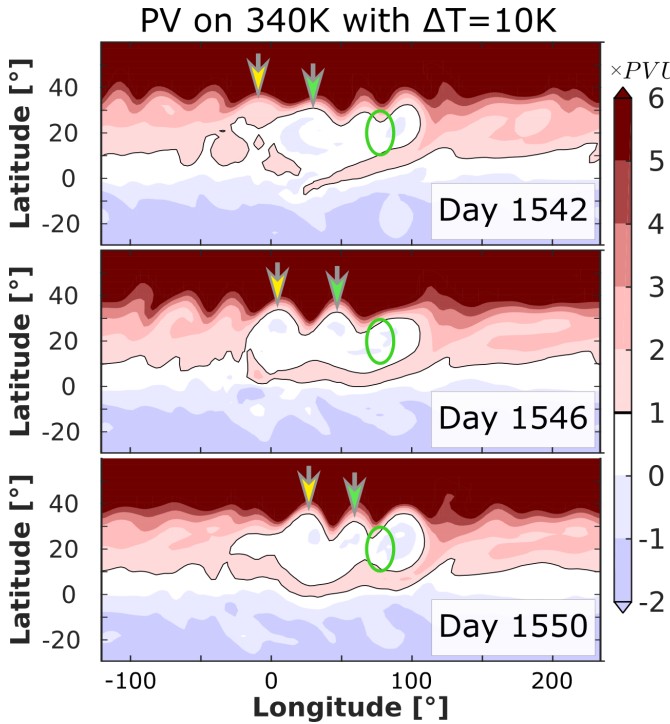

**Figure 16.** Snapshots of PV at 340 K for a forcing with $Q_0 = 5$ K/day and a symmetric basic state with $\Delta T = 10$ K. Ellipses indicate the forcing region and the 1 PVU contour is emphasised. The arrows illustrate the position of different wave structures on the sharp PV gradient associated with the jet stream.

Amemiya and Sato (2018) observed a somewhat similar behaviour in a steadily forced single-layer model with meridionally varying mean depth, inducing a background westerly mean wind. They describe a statistically steady localised response, which experiences westward shedding of eddies followed by a 'circling back' of shed vortices towards the forcing region. However, it should also be noted that the basic state used by Amemiya and Sato (2018) is associated with a pronounced reversal of the meridional PV gradient at the latitudes of the imposed forcing due to the varying mean depth of the modelled fluid layer. It is

questionable if such a state is a useful representations of the sub-tropical UTLS, as to some extent discussed by e.g. Kraucunas and Hartmann (2007), who argue that sub- and extratropical basic state flows should not be introduced through a layer depth gradient to avoid unrealistic behaviour of the system.

     The process shown in Figure 16 leads to the emergence of coherent PV anomaly structures at about -20° longitude and subsequent eastward propagation of such, as shown in Figure 14c. However, the meridional advection does not lead to a

complete separation of low-PV patches from the bulk anticyclone and subsequent eastward propagation of these patches, as we observe for basic state configuration with stronger meridional temperature gradient (see Figure 15). A potential explanation is





that the baroclinic eddies in the case with $\Delta T = 10$ K are too weak to produce sufficient meridional advection of PV and/or are located to close to the forcing region to create elongated PV filaments (recall that the jet core is located farther north for larger values of $\Delta T$).

Different authors have presented indication for the existence of a bi-modality of the monsoon anticyclone, with respective east- or westward displacement of the anticyclone centre (e.g., Zhang et al. (2002)). Similar theories suggest the occurrence of a 'split-state' of the anticyclone, showing two distinct centres (e.g., Vogel et al. (2015)). Nützel et al. (2016) found that only one (NCEP-1) out of seven analysed re-analysis datasets showed indication for a pronounced bi-modality for a range of time scales. The modulations of the PV low of the anticyclone by baroclinic eddies in Figure 16 can produce pronounced and zonally

separated PV minima on certain latitudes. It is not clear if this mechanism, however, can potentially explain the observation of mentioned split-phases in certain diagnostics and datasets.

Since the phenomenon described above modulates the northern edge of the anticyclone it creates a north-south asymmetry of the PV minimum in the form of a wavy northern edge and an only weakly perturbed southern edge. We find similar modulations of the northern edge in PV fields of re-analysis datasets, e.g. just before a westward shedding event on July 15th in Figure 10.

We also observe east- and westward shedding behaviour in experiments with an (hemispherically asymmetric) summer basic state, i.e. when choosing $T_{as} = 20$ K in Equation 1. Figure 17 shows the time evolution of PV along two latitude lines on the 335 K isentrope. Clear evidence for both, westward shedding at the latitudes of the forcing region and eastward shedding north of it, is visible. Westward shedding seems to occur much more frequently than eastward shedding and the shed vortices travel much farther. As described for the symmetric cases in Figure 14 the westward shedding seems to happen mostly on the latitude

of the forcing, while eastward shedding has a clear signature to the north of the forcing region. Both characteristics, the north-/southward shift of east-/westward shedding and the relatively more pronounced westward events, seem to be consistent with re-analysis data observations and our general understanding of these features (e.g., Popovic and Plumb, 2001; Enomoto et al., 2003; Vogel et al., 2014).

A crucial aspect of the relative importance of east-/westward shedding is potentially given by the strength and position of the

mid-latitude jet relative to the forcing region. As explained, Figure 14 shows the change from west- to predominantly eastward shedding as we increase $\Delta T$ in the (symmetric) annual-mean HS state. This induces (among other things) a strengthening of the jet and baroclinic eddies, both features are essential for the occurrence of eastward shedding process. Hence we find the described transition as the mid-latitude flow becomes stronger with increasing $\Delta T$. In the (asymmetric) summer case we find evidence for both types of shedding, although a clear dominance of westward shedding can be seen. Figure 17 indicates

generally weaker and poleward shifted jet for $T_{as} = 20$ K, compared to the symmetric case with $T_{as} = 0$. The existence of such intermediate states, showing signatures of east- and westward shedding, can potentially explain the dominance of the westward shedding process with respect to eastward shedding in the summer case.



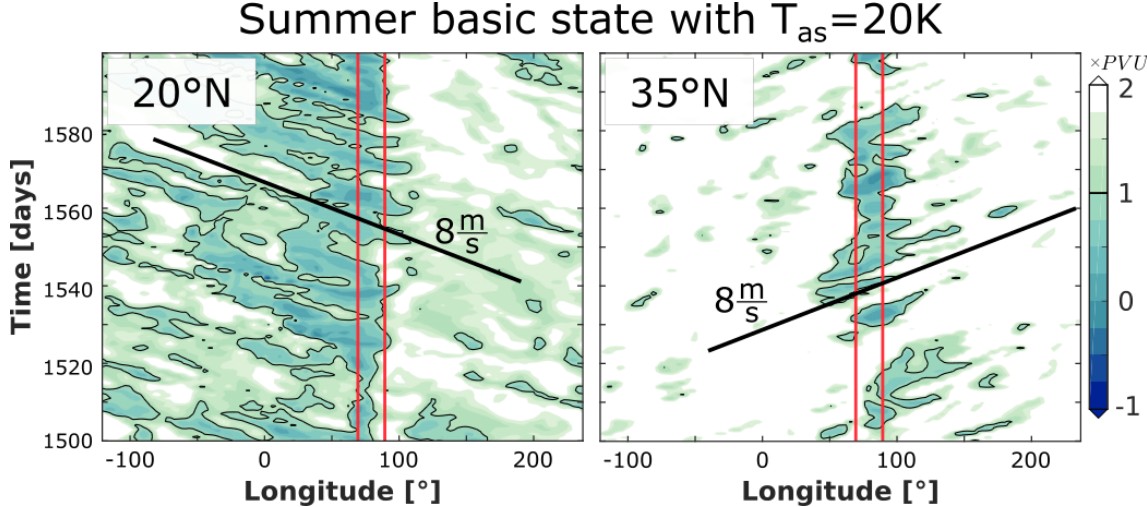

**Figure 17.** Time evolution of PV along two latitudes on the 335 K isentrope in a summer basic state with $\Delta T = 60$ K. The forcing magnitude is $Q_0 = 5$ K/day. Vertical lines show the extent of the forcing, the 1 PVU contour is highlighted. Black solid lines indicate an east-/westward velocity of 8 m/s.

## 5 Summary and conclusions

In this study we analysed the response of a three-dimensional dry dynamical model to a steady and spatially localised imposed
heating distribution, aimed to model the Asian monsoon anticyclone circulation. Particular focus was given on the modification
of the response when the localised forcing was applied to an atmosphere that includes a simple representation of mid-latitude
dynamics (mid-latitude jet and baroclinic eddies) compared to an atmosphere at rest. In a range of numerical experiments we
identified a set of characteristics and behaviours that are potentially relevant when describing the three-dimensional circulation
of the Asian monsoon anticyclone, including a localised zonal scale and eastward and westward eddy shedding phenomena.

As shown, in re-analysis data the time mean structure of the PV low associated with the monsoon anticyclone is zonally
localised and thus confined to the vicinity of the forcing region. In numerical model simulations with resting basic state ($\Delta T =
0$), however, the response to a simple monsoon heating is only very weakly localised, with the anticyclone extending far away
from the forcing region and its zonal extent essentially being determined by the weak thermal damping of the upper troposphere
lower stratosphere. In experiments with mid-latitude dynamics ($\Delta T \neq 0$) the response forms a localised anticyclone at the
latitudes of the forcing and a change in zonal mean flow to north of forcing region. Further examination supports the idea that
these two aspects of the response are effectively independent and that the interactions with mid-latitude dynamics, in particular
the baroclinic eddies, allow a time-mean response that is localised to the west and also extends to the (north-)east. The details
of the spatial structure of the anticyclone change as $\Delta T$ is varied or the basic state is changed from a more annual-mean state
to a state more representative of the summer time conditions in the northern hemisphere.





The dependence of the spatial structure of the time-mean response on the background is easier to understand if we look in more detail at time dependence of the system. In the case with resting basic state the response shows a transition from a steady beta-plume state to a state with westward eddy shedding for sufficiently strong forcing. In cases with time-varying basic state the nature of the time dependence of the response changes significantly and the system exhibits a transition from a state with westward shedding only to a state dominated by eastward shedding as $\Delta T$ increases, and thus structure and strength of

the background flow change. While in some cases (equinox flow) westward shedding seems to be inhibited, in other cases westward shedding persists (summer flow) and the response shows signs of both shedding-behaviours (as observed in the real atmosphere).

The presented model exhibits a range of behaviours and reproduces various properties of the monsoon circulation. The combined simplicity of the setup and ability to simulate different monsoon characteristics provides potential conceptual expla-

nations for many aspects of the monsoon structure and variability and gives a way to study them quantitatively.

*Author contributions.* PR produced the idealised model simulations, analysed the corresponding output, produced the visualisations and wrote the paper. PH advised PR throughout this work. contributed to the interpretation of the results and considerably improved the paper for the final version.

*Competing interests.* The authors declare that they have no conflict of interest.

*Acknowledgements.* This study was funded by the European project 'StratoClim' (7th Framework Programme, project no. 603557) and the European Research Council (ACCI, project no. 267760)



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
