# Peer review of "Zonal scale and temporal variability of the Asian monsoon anticyclone in an idealised numerical model"

_Weather and Climate Dynamics, 2020_

## Referee Comment (RC1)

*Zonal scale and temporal variability of the Asian monsoon anticyclone in an idealised numerical model* by Rupp & Haynes is a very clear investigation into the upper tropospheric Asian monsoon anticyclone. The spatial scale of this prominent summertime atmospheric feature has been a puzzle and it is nice to see a clear numerical experiment that shows the possible means by which longitudinal localization is achieved that does not involve *ad hoc* frictional or damping terms. The authors begin by showing that the response of the atmosphere to local heating with a base state of rest is almost unconfined longitudinally. Whereas, a base state that supports midlatitude jets and baroclinic eddies immediately leads to a PV anomaly that confined in the zonal direction. Further, the time dependent solution with differing forcing strengths shows a shedding of vortices from this anticyclone which have important implications for local monsoons. In particular, the change from eastward to westward shedding events depending on base state (and thus the midlatitude flow) is interesting and certainly worthy of further investigation. In all, I am happy to recommend the publication of this article in *Weather and Climate Dynamics*, but I have one or two main questions and some other points (detailed below) that the authors should try to address.

Main Comments

1. I see the remarkable change in the response of the atmosphere when the base state is changed. Specifically, the zonal restriction of the response when a state with jets and midlatitude baroclinic waves is considered, as opposed to the unconfined nature of the response with a base state of rest. But why? What is the physical reason for this change in the response? Are the baroclinic eddies acting as a means of eddy-diffusivity for the PV? Does this take the place of an *ad hoc* damping?

2. What sets the scale of the PV anomaly? Indeed, in figures 4,7 and 9 we see a zonally confined PV response with large $\Delta T$. In Figure 9 this is measured to be about 5,000 km. Is there an estimate where this length scale comes from? Is this a reflection of the scale of baroclinic eddies? Also, this scale is much smaller than the observed anticyclone that measures about 10,000 km in Figure 1. Any reasons for this mismatch?

3. The anticyclones obtained in these simulations all have a marked tilt, but the actual anticyclone in reanalysis (Figure 1) does not. Any thoughts on why this is the case?

Specific Comments

1. In these simulations, at early times, does the model produce wavetrains like those seen in typical equatorial heat source interacting with a jet type problems (for eg. those seen in Sardeshmukh & Hoskins 1988)?

2. In Figures 3, 7 & 9, it might be useful to include contours of the zonal mean flow.

3. Figures 1 and 10. Please either remove, or maintain, the aspect ratios of country outlines in these figures.

4. Line 315 and Figure 4. Does this "superposition" like result point to the linearity of the phenomenon?

5. Line 360 and Figure 7. Why does the PV anomaly become weaker with increasing $\Delta T$?

6. Line 405 and Figure 10f. Don't we also see a piece moving westward on Aug 1?

7. Figure 10. As an aside, this split of the anticyclone and its reformation is reminiscent of the break-up/reforming of the polar vortex! Any comments?

8. Line 440 and Figure 12. It's quite difficult to follow the strengthening and shedding of vortices in this figure. Won't a line plot at 20N of the be easier to follow? Further, what is going on the southern hemisphere? There seems to be a similar vortex formation and translation in the subtropics.

---

## Author Comment (AC1)

We thank the referee for carefully reading our manuscript, and for their constructive comments. In the following we will respond to the various comments and point out any changes we made to the paper based on them. Line numbers and figure references in the reviewer's comments refer to the original manuscript, while all references in the responses refer to the revised version of the manuscript. The comments of the referees are in black italics; our responses are in blue.

*Zonal scale and temporal variability of the Asian monsoon anticyclone in an idealised numerical model by Rupp & Haynes is a very clear investigation into the upper tropospheric Asian monsoon anticyclone. The spatial scale of this prominent summertime atmospheric feature has been a puzzle and it is nice to see a clear numerical experiment that shows the possible means by which longitudinal localization is achieved that does not involve ad hoc frictional or damping terms. The authors begin by showing that the response of the atmosphere to local heating with a base state of rest is almost unconfined longitudinally. Whereas, a base state that supports midlatitude jets and baroclinic eddies immediately leads to a PV anomaly that confined in the zonal direction. Further, the time dependent solution with differing forcing strengths shows a shedding of vortices from this anticyclone which have important implications for local monsoons. In particular, the change from eastward to westward shedding events depending on base state (and thus the midlatitude flow) is interesting and certainly worthy of further investigation. In all, I am happy to recommend the publication of this article in Weather and Climate Dynamics, but I have one or two main questions and some other points (detailed below) that the authors should try to address.*

*Main Comments*

1. *I see the remarkable change in the response of the atmosphere when the base state is changed. Specifically, the zonal restriction of the response when a state with jets and midlatitude baroclinic waves is considered, as opposed to the unconfined nature of the response with a base state of rest. But why? What is the physical reason for this change in the response? Are the baroclinic eddies acting as a means of eddy-diffusivity for the PV? Does this take the place of an ad hoc damping?*

We noted in the first version of the paper (in discussion of Figs. 4 and 8) that there likely to be two effects of changing the basic state by increasing ΔT. One is an advective effect, changing from a background state at rest to a background state with a strong westerly jet at midlatitudes in the upper troposphere, and the other is the stirring effect of the baroclinic eddies (we preferred simply to describe this as 'stirring' rather than to introduce the idea of a diffusivity). Because the increasing strength of the midlatitude jet and the increasing baroclinic eddy activity go together, it is difficult to distinguish between them. Therefore we see it as appropriate to mention both as contributing to this effect.

To make sure that this point is not missed -- it is indeed a conclusion of the paper that this combination of effects is important -- we have now reiterated the general point in the final 'Summary and Conclusions' section as well as adding it as part of the discussion of Figure 4.

2. *What sets the scale of the PV anomaly? Indeed, in figures 4,7 and 9 we see a zonally confined PV response with large ΔT. In Figure 9 this is measured to be about 5,000 km. Is there an estimate where this length scale comes from? Is this a reflection of the scale of baroclinic eddies?*

We noted in the paper that potential mechanisms to localise the response are advection and eddy stirring (as discussed above), together with thermal damping. It is the combined effect of these that sets the scale. Figure 7 shows that the scale shrinks as ΔT increases. We do not believe that the scale of the baroclinic eddies (in the sense of the scale of individual eddies)

is particularly important -- it is rather the overall stirring effect that is the relevant factor. We have added a corresponding note to the discussion of Figure 7.

> *Also, this scale is much smaller than the observed anticyclone that measures about 10,000 km in Figure 1. Any reasons for this mismatch?*

Our objective is to show that there is a strong dependence of the scale of the anticyclone on ΔT, rather than to imply that any particular value of ΔT gives the correct scale. These are highly idealised experiments that and it would not be surprising if not all quantitative features of flows in the real atmosphere could be matched for a single value of ΔT. We have modified Figure 9 and amended the discussion in Section 5 to make sure that this is clear.

> 3. *The anticyclones obtained in these simulations all have a marked tilt, but the actual anticyclone in reanalysis (Figure 1) does not. Any thoughts on why this is the case?*

Although it is not as pronounced as the tilt of the response in some of our experiments (e.g., Fig. 3d) the low-PV structure of the monsoon anticyclone in Fig. 1 shows signs of a tilting on its western and eastern flanks (e.g. the 4PVU contour between -50° and 50° longitude or the 4 and 5PVU contours around 150° longitude). Here again one needs to keep in mind the highly idealised nature of our experiments. Details of the background state, the structure of the imposed heating distribution and other aspects of the system (e.g., orographic forcing) will have a strong influence on the specifics of the structure of the response (and with it the prominence of the apparent tilt). We added a corresponding note to the discussion of Figure 9.

**Specific Comments**
> 1. *In these simulations, at early times, does the model produce wavetrains like those seen in typical equatorial heat source interacting with a jet type problems (for eg. those seen in Sardeshmukh & Hoskins 1988)?*

Certainly Rossby wave dynamics is contained within our model and we do see an identifiable Rossby wave response in some of our experiments, like the positive PV anomalies at -20° latitude in Fig. 11b (which do not correspond to closed contours in the full PV field). However, in making comparisons with pictures such as those in Sardeshmukh and Hoskins (1988) it is important to remember that those were generated in a single-layer model and that baroclinic instability was entirely absent. Indeed the work on 'deterministic' Rossby wave patterns carried out in the 1980s was important, but it made the gross simplification that the waves were imposed on a steady background, whereas for the real atmosphere a Rossby-wave response is a large-scale low-frequency response superimposed on a state that is highly variable in time as a result of baroclinic instability, with, potentially, non-trivial interactions between the baroclinic eddies and the larger scale Rossby waves. That has been an ongoing theme of work in this area over the last 20 years or so and we believe it is important to that investigations of the monsoon anticyclone include this ingredient. (Indeed our conclusion regarding the role of eddy stirring implies that it is essential to include the effect of the eddies in a description of the monsoon anticyclone.)

> 2. *In Figures 3, 7 & 9, it might be useful to include contours of the zonal mean flow.*

We thank the referee for this suggestion and have added zonal mean zonal wind profiles to some of the Figures to make it easier for the reader to see potential effects of advection by the mean background wind.

3. *Figures 1 and 10. Please either remove, or maintain, the aspect ratios of country outlines in these figures.*

Figure 1 and the different panels in Fig. 10 already have the same aspect ratio and show the same domain in terms of latitude and longitude, so no corresponding changes to the manuscript have been made.

4. *Line 315 and Figure 4. Does this "superposition" like result point to the linearity of the phenomenon?*

The results shown in Figure 4 do indicate that the two parts of the time-mean response (localised monsoon anticyclone and annular mode signal) can be regarded as rather independent of each other and individually as the time-mean responses to, respectively the zonal mean and zonally varying part of the forcing. This indeed implies some sort of additivity in the time-mean response (which is necessary for linearity but not sufficient -- we have not shown that the individual responses scale linear with the amplitudes of the different parts of the forcing). There may well be linearity, but one has to be careful how this is interpreted and how the term 'linear' is used. What is being considered here is the time-mean response of a system that includes nonlinear interactions between zonal mean flow, large-scale waves and synoptic-scale eddies. The linearity is in the operator that incorporates these interactions in predicting the time-mean response to a forcing. 'Linearity' does not imply, for example, that 'linear wave theory with a fixed background state' is a predictor of the response.

5. *Line 360 and Figure 7. Why does the PV anomaly become weaker with increasing T?*

The weakening of the PV anomaly seen in Fig. 7 with increasing $\Delta T$ can potentially be explained by an increasing strength of the baroclinic eddy field of the basic state and an increasing importance of the corresponding stirring effect on the flow forced by the imposed heating. This is further indication for the potential importance of interactions between the response to the imposed heating and mid-latitude eddies. We added a corresponding note to the discussion of Figure 7.

6. *Line 405 and Figure 10f. Don't we also see a piece moving westward on Aug 1?*

The slight westward propagation/extension of the PV low of the bulk anticyclone (with 2PVU contour located at about 30°N and 60°W on July 30th and at about 40°W on August 1st) could be a sign of the onset of another westward shedding event, although the high internal variability of the system (e.g., due to interactions of the monsoon anticyclone with the baroclinic eddy field) make it difficult to interpret such details of the flow evolution. We have revised some of the discussion of Figure 10.

7. *Figure 10. As an aside, this split of the anticyclone and its reformation is reminiscent of the breakup/reforming of the polar vortex! Any comments?*

As both, the monsoon anticyclone and the polar vortex, can be described as coherent vortices defined by closed PV contours their fundamental dynamics will certainly show some degree of similarity. However, the processes that lead to the split of the monsoon anticyclone (in the context of eddy shedding) seen in Fig. 10 and the break up of the polar vortex (e.g. during a sudden stratospheric warming) are fundamentally different. The phenomena of eddy shedding are caused by the internal dynamics of the system (westward shedding) or by interactions with horizontally propagating synoptic scale waves (eastward shedding), while sudden warmings are essentially caused by vertical propagation and dissipation of planetary scale waves.

*8. Line 440 and Figure 12. It's quite difficult to follow the strengthening and shedding of vortices in this figure. Won't a line plot at 20N of the be easier to follow? Further, what is going on the southern hemisphere? There seems to be a similar vortex formation and translation in the subtropics.*

We agree that in a line plot it is easier to see the growing of distinct vortices within the forcing region and the subsequent westward shedding, which is the crucial aspect of Figure 12. We substituted the 2D contour plot showing stream function at 13km by a corresponding line plot of stream function at 13km and 20° latitude (both labelled Fig. 12 in the originally submitted and revised version of the manuscript, respectively).

Regarding the negative stream function signal in the southern hemisphere in Figure 12 (also visible in e.g. Fig. 11b), this is likely part of the Rossby wave response of the atmosphere to the localised forcing. These structures do not correspond to closed PV contours and should therefore not directly be regarded as discrete vortices. We added a corresponding note to the discussion of Fig. 11 clarifying the potential nature and significance of these structures.

---

## Author Comment (AC2)

We thank the referee for carefully reading our manuscript, and for their constructive comments. In the following we will respond to the various comments and point out any changes we made to the paper based on them. Line numbers and figure references in the reviewer's comments refer to the original manuscript. The comments of the referees are in black italics; our responses are in blue.

*This paper describes an idealized model of the Asian monsoon anticyclone using a dry dynamical core model, to analyze the interactions of localized monsoon forcing and midlatitude baroclinic eddies. The paper focuses on understanding the model sensitivity to monsoon strength and imposed background circulation, and the results nicely highlight the region of active dynamics on the poleward side of the anticyclone. The key points include that the midlatitude eddy interactions lead to a zonal localization of the monsoon circulation, and eastward/westward eddy shedding occurs spontaneously in the model depending on parameter settings. The zonal localization from eddy interactions helps address the issue of overly strong dissipation required in previous idealized models, and the sensitivity of the model eddy shedding to model parameters helps identify the underlying processes in a simplified setting. The paper does a good job of placing this work in contest of previous modeling work and observations. The results are interesting and novel and the paper is well written; I enjoyed reading it very much. I have only minor comments for the authors to consider in revision.*

*I suggest including a few additional recent references and related discussions:*

*Sui and Bowman (2019) regarding monsoon modeling DOI:10.1175/JAS-D-18-0340.1*
*Siu and Bowman (2020) for analysis of sub-vortices and eddy shedding 10.1175/JASD-19-0349.1*
*Honomichl and Pan (2020) for analysis of eddy shedding 10.1029/2019JD032094*

We thank the referee for pointing out these recent studies related to the monsoon anticyclone. We added corresponding references to the manuscript.

*Line 353: Fig. 5 should be Fig. 6*

This was corrected.

*The sensitivity to anticyclone forcing demonstrated in Fig. 11 is nice. What controls the westward phase speed of the eddies for the Q=5 K/day case? Are these Rossby waves?*

The structures seen at 20° latitude in Figures 11b and d are distinct vortices and correspond to closed PV contours. Within the framework of single-layer beta-plane models it has been shown that isolated vortices propagate to leading order with the long Rossby wave speed, which (in a barotropic and quasi-geostrophic system) is given by $c=\beta R_D^2$, where $\beta$ is the meridional gradient of planetary vorticity and $R_D$ is the Rossby radius of deformation. For our system (subtropical upper troposphere with $\beta \approx 2*10^{-11}s^{-1}$, $R_D \approx 10^6$m) we find $c \approx 20$m/s, which is of the same order as the eddy velocity shown in Fig. 11d. We added a corresponding statement to the figure description including a reference to the Rupp PhD thesis with a more detailed discussion of the problem. We further added a note to emphasise that these structures are indeed coherent vortices and not Rossby waves, as they correspond to closed contours in the full PV field.

*What is going on with the traveling eddies centered near 15 S (also in Fig. 12)?*

These structures do not correspond to closed contours in the full PV field and are likely to be essentially a Rossby wave response forced by the eddy shedding and eddy propagation process. We added a corresponding note to the discussion of Figure 11.

*Line 525 and following: it is useful to cite the more recent analyses of Siu and Bowman (2020) and Honomichl and Pan (2020) regarding observations of sub-vortices, eddy shedding and 'bi-modality'.*

We added a corresponding reference to the paragraph.